# Leveraging Variable Sparsity to Refine Pareto Stationarity in Multi-Objective Optimization

**Zeou Hu**

University of Waterloo, Vector Institute

`zeou.hu@uwaterloo.ca`

**Yaoliang Yu**

University of Waterloo, Vector Institute

`yaoliang.yu@uwaterloo.ca`

## Abstract

Gradient-based multi-objective optimization (MOO) is essential in modern machine learning, with applications in e.g., multi-task learning, federated learning, algorithmic fairness and reinforcement learning. In this work, we first reveal some limitations of Pareto stationarity, a widely accepted first-order condition for Pareto optimality, in the presence of sparse function-variable structures. Next, to account for such sparsity, we propose a novel solution concept termed Refined Pareto Stationarity (RPS), which we prove is always sandwiched between Pareto optimality and Pareto stationarity. We give an efficient partitioning algorithm to automatically mine the function-variable dependency and substantially trim non-optimal Pareto stationary solutions. Then, we show that gradient-based descent algorithms in MOO can be enhanced with our refined partitioning. In particular, we propose Multiple Gradient Descent Algorithm with Refined Partition (RP-MGDA) as an example method that converges to RPS, while still enjoying a similar per-step complexity and convergence rate. Lastly, we validate our approach through experiments on both synthetic examples and realistic application scenarios where distinct function-variable dependency structures appear. Our results highlight the importance of exploiting function-variable structure in gradient-based MOO, and provide a seamless enhancement to existing approaches.

## 1 Introduction

Multi-objective optimization (MOO) aims to optimize several objective functions simultaneously over shared parameters, seeking solutions where no objective can be improved without worsening another. Many modern machine learning applications naturally involve multiple objectives. For example, in fair machine learning, one needs to balance model utility with various fairness notions, while in multi-task learning, models are trained across different tasks with shared representations. Similarly, federated learning and reinforcement learning often involve optimizing the performance of multiple agents or entities distributed across environments. The solutions to such problems are known as Pareto optimal (PO) solutions, representing a balance across the competing objectives. However, finding PO solutions becomes challenging in complex (*e.g.* high-dimensional) problems.

Among various approaches for solving MOO problems, one that has gained significant attention recently is the Multiple Gradient Descent Algorithm (MGDA) (Mukai, 1980; Fliege and Svaiter, 2000; Désidéri, 2012). MGDA extends classical gradient descent to optimize multiple objectives simultaneously. Unlike traditional linear scalarization, which requires predetermined weights for each objective, MGDA directly optimizes the vector-valued problem without such weighting. At each iteration, MGDA identifies a common descent direction that improves all objectives, a feature not guaranteed by scalarization methods. MGDA's ability to converge to a Pareto stationary solution, which serves as a first-order necessary condition for Pareto optimality, has made it widely applicable in areas like multi-task learning (Sener and Koltun, 2018), federated learning (Hu et al., 2022), and generative modeling (Albuquerque et al., 2019). While Pareto stationarity provides a solid theoretical foundation (as shown in Lemma 1 below), it can fall short in complex, especially sparse settings, creating room for a more refined approach, as we illustrate in Section 4.

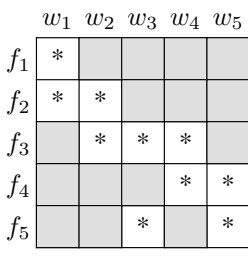 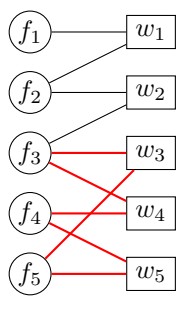 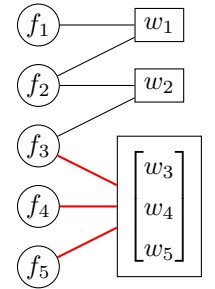 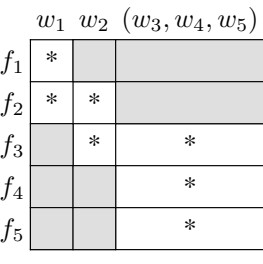

Figure 1: A simple illustration of our main idea. From left to right: adjacency matrix showing the function-variable dependency structure; the underlying bipartite graph with a cycle highlighted in red; merging the variables in the cycle to a subset and contracting the graph; adjacency matrix showing the new dependency structure. The process is iterated until the graph becomes acyclic.

In this work, we reveal that Pareto stationarity (PS), while necessary for Pareto optimality, is often insufficient. Even in convex settings, PS can frequently lead to sub-optimal PO solutions, especially when sparse function-variable dependencies cause strict convexity to fail. To address this, we propose Refined Pareto Stationarity (RPS), which leverages sparsity to identify a valid partition of variables and refines PS with respect to this partition. We prove that RPS is (strictly) sandwiched between PS and Pareto optimality (PO), making it a more useful and practical goal. Additionally, under *partial* strict convexity, we prove RPS reduces exactly to Pareto optimality, whereas the widely-used PS does not. These advantages are illustrated through intuitive examples and validated experimentally.

Since RPS is defined with respect to a refined partition, we propose an efficient partitioning procedure (REFINED_PARTITION) to automatically discover, if needed, the correct refined partition of variables, based on an input adjacency matrix encoding function-variable dependencies. Figure 1 illustrates our main idea through a simple example. More importantly, our results enable practitioners to enhance existing gradient-based descent algorithms for MOO by incorporating this partitioning routine and the refined stationary condition RPS. Specifically, we introduce Multiple Gradient Descent Algorithm with Refined Partition (RP-MGDA) to exploit these advances and we formally prove its convergence.

We summarize our contributions below, with further context provided in the next section:

- We present illustrative examples to reveal some limitations of Pareto stationarity, and motivate the need for a *valid* partitioning of variables when function-variable dependency is sparse.

- We propose RPS, a novel solution concept that refines Pareto stationarity, together with an efficient partitioning procedure REFINED_PARTITION to automatically discover the refined partition.

- We introduce RP-MGDA, an example of enhancing gradient-based descent algorithms in MOO by incorporating refined partitioning. We establish the convergence and complexity of RP-MGDA.

- We validate the effectiveness of our approach through experiments on synthetic and benchmark datasets, showcasing its application to ML scenarios with diverse dependency structures.

Our code is available at https://github.com/watml/RP-MGDA.

## 2 RELATED WORKS

Multi-objective optimization (MOO) and Pareto solution have been extensively studied, with traditional approaches such as evolutionary algorithms (Deb et al., 2002). However, these techniques face challenges when applied to modern machine learning problems, which are often better addressed by gradient-based methods. Consequently, in this work we focus on gradient-based MOO.

**Gradient-based MOO.** Gradient-based MOO involves optimizing multiple objectives using gradient information. A foundational algorithm in this regard is MGDA (Mukai, 1980; Fliege and Svaiter, 2000; Désidéri, 2012), which computes a common descent direction for all objectives by solving for the minimum-norm vector in the convex hull of gradients. Fliege et al. (2019) offered a detailed convergence analysis of MGDA, while subsequent works have also extended classical single-objective

methods to the multi-objective setting, including Newton's method (Fliege et al., 2009), (accelerated) proximal gradient (Tanabe et al., 2019; Tanabe et al., 2023), conditional gradient (Assunção et al., 2021), and subgradient method (O. Montonen and Mäkelä, 2018), among others. Another important line of research focuses on stochastic variants of MGDA, e.g. Mercier et al. (2018), due to their practical relevance in machine learning, particularly for mini-batch training of deep neural networks. Liu and Vicente (2021) introduced SMG as a direct stochastic counterpart of MGDA, albeit with a biased estimate of the common descent direction. Subsequent works of Zhou et al. (2022) and Fernando et al. (2023) track historical information to reduce bias, while Chen et al. (2023) and Xiao et al. (2023) propose unbiased estimate of the update direction through double sampling.

**MOO in Multi-task Learning (MTL).** MTL aims to train a common model that can solve multiple, potentially related tasks. Sener and Koltun (2018) first framed MTL as a multi-objective optimization problem and addressed it using MGDA. Since then, numerous works in MTL have approached this problem within the general MOO framework, treating task losses as objectives and proposing novel gradient aggregation schemes to mitigate task conflicts while addressing various aspects of MTL (e.g., Lin et al., 2019). PCGrad (Yu et al., 2020) projects each gradient onto the normal plane of other conflicting gradients and aggregates them. CAGrad (Liu et al., 2021a) balances minimizing the average task loss and improving the worst-performing objective by constraining the search region for the common direction in MGDA. Nash-MTL (Navon et al., 2022) formulates MTL as a bargaining game to weigh the gradients. I-MTL (Liu et al., 2021b) separately searches for the magnitude and direction of the common gradient. FairGrad (Ban and Ji, 2024) proposes a new gradient aggregation scheme to ensure $\alpha$-fairness in MTL.

**MOO in Fair machine learning (FML).** FML is inherently a multi-objective problem, due to the trade-off between utility and various notions of fairness that need not be compatible with each other (Kleinberg et al., 2017). A common strategy in FML is to linearly combine the multiple objectives through scalar weights (Kamishima et al., 2011), corresponding to the well-known scalarization technique in MOO, while later work, such as Padh et al. (2021), applies MGDA to handle these competing objectives more directly.

The works mentioned above either 1) adopt the general MOO formulation that treats all learnable parameters as a single variable $\mathbf{w} \in \mathbb{R}^d$, thereby disregarding any underlying function-variable structure, or 2) are confined to the MTL setting presented in Sener and Koltun (2018), whose variable partition scheme can be viewed as a special case of our more general framework. To our best knowledge, our work is the first to systematically investigate and refine Pareto stationary solutions in gradient-based MOO, by taking into account the function-variable structure. Our results are complementary to the above approaches and potentially integrable with many of them.

## 3 PRELIMINARIES

In this section, we briefly review the concepts of Pareto optimality and Pareto stationarity in multi-objective optimization. We then introduce the widely used MGDA method.

### 3.1 MULTI-OBJECTIVE MINIMIZATION (MOO)

In mathematical terms, a Multi-Objective Minimization (MOO) problem can be written as[1]

$$\min_{\mathbf{w} \in \mathbb{R}^d} \mathbf{f}(\mathbf{w}), \text{ where } \mathbf{f}(\mathbf{w}) := (f_1(\mathbf{w}), f_2(\mathbf{w}), \ldots, f_m(\mathbf{w})) \tag{1}$$

and the minimum is defined w.r.t. the *partial* ordering[2]:

$$\mathbf{f}(\mathbf{w}) \leq \mathbf{f}(\mathbf{z}) \iff \forall i = 1, \ldots, m, \; f_i(\mathbf{w}) \leq f_i(\mathbf{z}). \tag{2}$$

Unlike single objective optimization, with multiple objectives it is possible that

$$\mathbf{f}(\mathbf{w}) \not\leq \mathbf{f}(\mathbf{z}) \text{ and } \mathbf{f}(\mathbf{z}) \not\leq \mathbf{f}(\mathbf{w}), \tag{3}$$

in which case we say $\mathbf{w}$ and $\mathbf{z}$ are not comparable. As a result, there is usually a set of solutions that are all optimal (*a.k.a. Pareto Optimal*) for a given MOO problem, whose objective values form the *Pareto front*.

---

[1]We focus on minimization in this work, while maximization can be treated similarly.

[2]We remind that algebraic operations such as $\leq$ and $+$, when applied to a vector with another vector *or scalar*, are always performed component-wise.

## 3.2 Pareto Optimality and Pareto Stationarity

**Definition 1** (Pareto Optimality). *We call $\mathbf{w}_*$ a Pareto optimal solution of* (1) *if its objective value $\mathbf{f}(\mathbf{w}_*)$ is a minimum element w.r.t. the partial ordering in* (2)*; equivalently,*

$$\forall \mathbf{w}, \; \mathbf{f}(\mathbf{w}) \le \mathbf{f}(\mathbf{w}_*) \implies \mathbf{f}(\mathbf{w}) = \mathbf{f}(\mathbf{w}_*). \tag{4}$$

In other words, it is not possible to improve *any* component objective in $\mathbf{f}(\mathbf{w}_*)$ without compromising *some* other objective.

**Definition 2** (Weakly Pareto Optimality). *We call $\mathbf{w}_*$ a weakly Pareto optimal solution if there does not exist any $\mathbf{w}$ such that $\mathbf{f}(\mathbf{w}) < \mathbf{f}(\mathbf{w}_*)$, i.e., it is not possible to improve all component objectives in $\mathbf{f}(\mathbf{w}_*)$ simultaneously.*

Clearly, any Pareto optimal solution is also weakly Pareto optimal but the converse may not hold.

Next, we recall the concept of *Pareto Sationarity* (also referred to as *Pareto Criticality*), which is the first order necessary condition for Pareto optimality.

**Definition 3** (Pareto Stationarity). *We call $\mathbf{w}_*$ Pareto stationary iff*

$$\mathbf{0} \in \mathrm{conv}\{\nabla f_1(\mathbf{w}_*), \cdots, \nabla f_m(\mathbf{w}_*)\}, \tag{5}$$

i.e., *there exists some $\boldsymbol{\lambda} \ge 0$ such that $\sum_i \lambda_i = 1$ and $\sum_i \lambda_i \nabla f_i(\mathbf{w}^*) = \mathbf{0}$.*

The relevance of Pareto stationarity is captured in the following lemma:

**Lemma 1** (*e.g.*, Mukai, 1980, Theorem 1). *Any Pareto optimal solution is Pareto stationary. Conversely, if all functions are convex (resp., strictly convex), then any Pareto stationary solution is weakly Pareto optimal (resp., Pareto optimal).*

## 3.3 Multiple Gradient Descent Algorithm (MGDA)

MGDA has been independently proposed by Mukai (1980) and Fliege and Svaiter (2000) and Désidéri (2012) as a gradient-based method to find Pareto stationary solutions of an MOO problem. It has gained notable attention in machine learning in recent years, largely because of its gradient-based nature, in contrast to traditional MOO techniques. Compared with the scalarization technique (*i.e.*, linearly combining different objectives), MGDA removes the need to pre-determine any weight vector. More importantly, MGDA is guaranteed to improve all objectives of any starting solution while scalarization cannot (Mukai, 1980).

In each iteration, MGDA seeks a new solution that minimizes the maximum change among all objectives, i.e.,

$$\tilde{\mathbf{w}}^{t+1} = \underset{\mathbf{w}}{\mathrm{argmin}} \; \underset{\boldsymbol{\lambda} \in \Delta}{\max} \; \boldsymbol{\lambda}^\top (\mathbf{f}(\mathbf{w}) - \mathbf{f}(\mathbf{w}^t)), \tag{6}$$

where $\Delta$ is the standard simplex. Next, we upper bound $\mathbf{f}(\mathbf{w})$ by the usual quadratic approximation:

$$\underset{\mathbf{w}}{\min} \; \underset{\boldsymbol{\lambda} \in \Delta}{\max} \; \boldsymbol{\lambda}^\top J_{\mathbf{f}}^\top(\mathbf{w}^t)(\mathbf{w} - \mathbf{w}^t) + \tfrac{1}{2\eta}\|\mathbf{w} - \mathbf{w}^t\|^2. \tag{7}$$

where $J_{\mathbf{f}} = [\nabla f_1, \dots, \nabla f_m] \in \mathbb{R}^{d \times m}$ is the Jacobian and $\eta > 0$ is the step size. This approximation (7) is (strongly) convex in $\mathbf{w}$ (even for nonconvex $\mathbf{f}$) and concave in $\boldsymbol{\lambda}$, and hence strong duality holds. Swapping $\min$ with $\max$ in (7) and setting the derivative w.r.t. $\mathbf{w}$ to $\mathbf{0}$, we deduce the following "minimum-norm" direction:

$$\mathbf{d}^t = J_{\mathbf{f}}(\mathbf{w}^t)\boldsymbol{\lambda}_*^t, \;\; \text{where} \;\; \boldsymbol{\lambda}_*^t = \underset{\boldsymbol{\lambda} \in \Delta}{\mathrm{argmin}} \; \|J_{\mathbf{f}}(\mathbf{w}^t)\boldsymbol{\lambda}\|^2. \tag{8}$$

MGDA then performs the update along the direction $\mathbf{d}^t$:

$$\mathbf{w}^{t+1} = \mathbf{w}^t - \eta \mathbf{d}^t. \tag{9}$$

Common stopping criteria include $\|\mathbf{d}^t\| \le \mathtt{tol}$ for some tolerance $\mathtt{tol}$ or by capping at a maximum number of iterations.

When the step size $\eta$ is sufficiently small (so that (7) is indeed an upper bound on (6)), the MGDA update (9) *simultaneously* decreases all objectives (seen easily by the feasible choice $\mathbf{w} = \mathbf{w}_t$ in (7)). It is clear that the fixed points of MGDA are exactly Pareto stationary solutions. In fact, Fliege et al. (2019) proved that the minimum norm $\|\mathbf{d}^t\|$ converges to 0 at rate $O(1/\sqrt{t})$, matching the well-known result in single objective optimization.

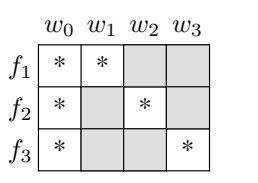 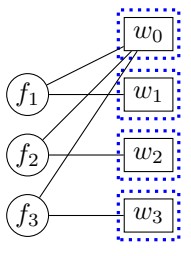 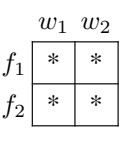 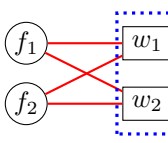

Figure 2: Illustration of Example 1 and Example 2. Left: function-variable dependency structure of Example 1. There is no cycle in the underlying bipartite graph so our method partitions each variable into its own block. Right: function-variable dependency structure of Example 2. There is a cycle (highlighted in red) in the underlying bipartite graph so our method merges both variables into a single block. In both examples, MGDA treats all variables as a single block.

## 4  ILLUSTRATIVE EXAMPLES

We construct two examples that illustrate the various definitions above and reveal some subtle issues about Pareto stationarity and the (coordinate-wise) MGDA update.

**Example 1.** *Let $f_1(\mathbf{w}) = w_0^2 + w_1^2$, $f_2(\mathbf{w}) = w_0^2 + w_2^2$ and $f_3(\mathbf{w}) = w_0^2 + w_3^2$. Clearly, all component functions are convex. There is a unique Pareto optimal solution $\mathbf{w}_\star = (0, 0, 0, 0)$, but there are infinitely many Pareto stationary (in fact, weakly Pareto optimal) solutions:*

$$\{(0, 0, w, z), (0, w, 0, z), (0, w, z, 0) : w, z \in \mathbb{R}\}. \tag{10}$$

*It is clear the MGDA update* (8)-(9) *could get trapped at any of the Pareto stationary solutions. However, if we update the variables separately, the coordinate-wise MGDA converges to the unique Pareto optimal solution. See Figure 2 for a pictorial illustration.*

We note that Example 1 does not contradict Lemma 1, since none of the component functions is *strictly* convex. In general, strict convexity, or even convexity, is rarely satisfied in deep learning.

It is then tempting to *always* partition the variables and update them separately using MGDA (*i.e.*, a coordinate-wise MGDA). However, the following example shows this idea may also lead us astray.

**Example 2.** *Let $f_1(\mathbf{w}) = (w_1 - w_2)^2 + w_1^2$ and $f_2(\mathbf{w}) = (w_1 - w_2)^2 + w_2^2$. Clearly, both functions are strictly convex. There is a unique Pareto stationary (and hence Pareto optimal) solution $\mathbf{w}_\star = (0, 0)$. However, if we apply MGDA to each variable separately, it may converge to any point on the diagonal $\{(w, w) : w \in \mathbb{R}\}$. On the other hand, the vanilla MGDA (treating both variables as a whole) converges to the unique Pareto stationary solution. See Figure 2 for a pictorial illustration.*

In the next section, by leveraging on the (sparse) function-variable dependency structure, we prove that it is possible to identify a *valid* partition of variables and define refined Pareto stationary solutions that are sandwiched by Pareto optimal solutions and Pareto stationary solutions.

## 5  REFINED PARETO STATIONARY SOLUTIONS

Motivated by the aforementioned examples, we now consider partitioning the variables in the MOO problem (1). Let $\mathcal{P} = \{P_1, \ldots, P_k\}$ be a partition of $[d] := \{1, \ldots, d\}$. For a subset $P$ of variables, we use $\mathbf{f}^P$ to denote the set of functions $f_i$ that depend on some variable in $P$.

**Definition 4** (Generalized Pareto Stationarity). *We call $\mathbf{w}_*$ Pareto stationary w.r.t. the partition $\mathcal{P}$ iff*

$$\mathbf{0} \in \prod_{j=1}^{k} \mathrm{conv}\{\nabla_{\mathbf{w}_{P_j}} \mathbf{f}^{P_j}(\mathbf{w}_*)\} \ (\textit{the Cartesian product}), \tag{11}$$

*i.e., for all $j \in [k]$, there exists $\boldsymbol{\lambda}^j \in \Delta$ such that $\sum_i \lambda_i^j \nabla_{\mathbf{w}_{P_j}} f_i(\mathbf{w}^*) = \mathbf{0}$ and $\lambda_i^j = 0$ if $f_i$ does not depend on the variables in $P_j$.*

When $\mathcal{P} = [d]$ is the trivial partition, our definition reduces to Pareto stationarity in Definition 3. The following lemma slightly generalizes one direction of Lemma 1.

**Lemma 2.** *Any Pareto optimal solution is Pareto stationary w.r.t. any partition.*

(All proofs can be found in Appendix A.) However, the converse part of Lemma 2 under (strict) convexity is harder to extend: we need to find the right partition first. Indeed, Example 2 already showed Pareto stationarity w.r.t. the finest partition is not equal to Pareto optimality, or put it slightly differently, Pareto stationary w.r.t. an arbitrary *finer* partition does not necessarily imply Pareto stationary w.r.t. a *coarser* partition[3].

Consider the adjacency matrix $A \in \{0, 1\}^{m \times d}$ of a bipartite graph, with

$$A_{ij} = \begin{cases} 1, & \text{if } f_i \text{ depends on } w_j \\ 0, & \text{otherwise} \end{cases}. \tag{12}$$

Our next goal is to find the (finest) partition $\mathcal{P}$ of variables (*i.e.*, columns of $A$) so that Pareto stationary w.r.t. $\mathcal{P}$ is also Pareto stationary (w.r.t. the trivial partition). This is particularly useful when the dependency matrix $A$ is sparse, since it enables us to largely reduce the set of Pareto stationary solutions (without missing any Pareto optimal ones); see Example 1.

The main idea is to find cycles in the underlying bipartite graph, merge and contract the resulting variables, and repeat until the graph becomes acyclic; see Figure 1 for an illustration and Appendix A.1 for pseudo-code. We recall from Cormen et al. (2009, Exercise 22.4-3, p. 615) that detecting a cycle in an undirected graph using depth-first search costs $O(m + d)$ (the number of nodes) for our bipartite graph. Since merging the variables in a cycle reduces the graph size by at least 1, the process can repeat at most $O(d)$ times. Therefore, the entire process, which we denote as REFINED_PARTITION $(A)$, costs $O(d(m + d))$ to output a partition $\mathcal{P} := \{P_1, \dots, P_d\}$ of the variables.

We formally justify the above partition procedure in:

**Theorem 1.** *Let $\mathcal{P} =$ REFINED_PARTITION$(A)$ where $A$ is defined in* (12)*. Then, any $\mathbf{w}_*$ that is Pareto stationary w.r.t. $\mathcal{P}$, which we call refined Pareto stationary from now, is Pareto stationary.*

We are finally ready to provide a converse of Lemma 2, under (partial) strict convexity.

**Theorem 2.** *Let $\mathcal{P} = \{P_1, \dots, P_k\}$ be returned by* REFINED_PARTITION*. Suppose each $f_i$ is strictly convex in $\mathbf{w}_{I_i}$ where $I_i \subseteq \mathcal{P}$ is the set of blocks of variables that $f_i$ depends on. Then, any refined Pareto stationary solution (w.r.t. $\mathcal{P}$) is Pareto optimal.*

To summarize, refined Pareto stationarity (based on our partition procedure) is sandwiched between Pareto optimality and Pareto stationarity; Figure 3 illustrates the relationships between these concepts. In particular, we have identified a *valid* partition to sharpen Lemma 1 in its full extent. Importantly, applying our refined Pareto stationarity to Example 1 and Example 2 successfully identifies the unique Pareto optimal solution while eliminating infinitely many spurious Pareto stationary solutions.

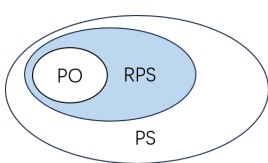

Figure 3: Euler diagram for Pareto Optimal (PO), Refined Pareto Stationary (RPS) and Pareto Stationary (PS).

## 6 IMPROVING EXISTING ALGORITHMS WITH THE REFINED PARTITION

The refined Pareto stationarity introduced in the previous section is a solution concept. Broadly, it can be used to enhance existing gradient-based *descent* algorithms for MOO. Indeed, we can simply cycle through or randomly pick the blocks of variables identified by REFINED_PARTITION, and update each block with a chosen gradient-based descent algorithm in turn. The resulting algorithm is guaranteed to be descending, *i.e.*, it improves all objectives in each step.

Interestingly, for certain MOO algorithms we can even afford to update all blocks *simultaneously*. We demonstrate this possibility and its practical value by modifying and analyzing the MGDA algorithm from Section 3.3, enabling it to converge to a refined Pareto stationary solution. The full algorithm, which we call RP-MGDA, is summarized in Algorithm 1. The algorithm again terminates when $\|\mathbf{d}\|_2 < \texttt{tol}$, or when a maximum number of iterations is achieved.

---

[3]The subtlety lies on the requirement that each $\boldsymbol{\lambda}^j \in \Delta$, *i.e.*, we cannot set any of them to $\mathbf{0}$.

---

**Algorithm 1:** Multiple Gradient Descent Algorithm with Refined Partition (RP-MGDA)

---

**Input:** function-variable dependency structure $A$, functions $\mathbf{f}$, initializer $\mathbf{w} \in \mathbb{R}^d$, learning rate $\boldsymbol{\eta}$

1   $(P_1, \ldots, P_k) = \text{REFINED\_PARTITION}(A)$    // partition variables based on $A$
2   **for** $t = 1, 2, \ldots$ **do**
3     **parfor** $j = 1, 2, \ldots, k$ **do**                             // in parallel
4       $J^j(\mathbf{w}) \leftarrow \nabla_{\mathbf{w}_{P_j}} \mathbf{f}^{P_j}(\mathbf{w})$                // compute gradients
5       $\boldsymbol{\lambda}^j \leftarrow \text{argmin}_{\boldsymbol{\lambda} \in \Delta} \|J^j(\mathbf{w})\boldsymbol{\lambda}\|^2$         // solve the dual subproblem
6       $\mathbf{d}^t_{P_j} \leftarrow J^j(\mathbf{w})\boldsymbol{\lambda}^j$          // minimum-norm direction in $P_j$
7       $\mathbf{w}_{P_j} \leftarrow \mathbf{w}_{P_j} - \eta_j \cdot \mathbf{d}^t_{P_j}$                    // update

**Output:** final weights $\mathbf{w}$

---

Following Fliege et al. (2019) we prove that the direction $\mathbf{d}$ in RP-MGDA (Algorithm 1) converges to $\mathbf{0}$ at the rate $O(1/\sqrt{t})$, and hence matching the usual result for single objective optimization.

**Theorem 3.** *Suppose the functions $f_i$ are bounded from below and Lipschitz smooth with constant $L_i$. Set $\eta_j \equiv \eta \leq \min_i \frac{1}{L_i}$. Then, $\min_{1 \leq t \leq T} \|\mathbf{d}^t\| \leq O(\frac{1}{\sqrt{T}})$.*

Combining Theorem 1 and Theorem 3 we immediately have the following result:

**Corollary 1.** *Assuming RP-MGDA converges, then it converges to a Pareto stationary solution at the rate $O(1/\sqrt{t})$.*

A few remarks regarding RP-MGDA are in order: **1)** When the sublevel set $\{\mathbf{w} : \mathbf{f}(\mathbf{w}) \leq \mathbf{f}(\mathbf{w}^1)\}$ is compact, RP-MGDA admits a limit point that is refined Pareto stationary and hence Pareto stationary, thanks to its descending property. **2)** In practice we do not need to know the Lipschitz constants $L_i$: a standard line search procedure on $\eta$ to test the quadratic upper bound (7) suffices to achieve the same convergence rate. **3)** Theorem 1 and hence Corollary 1 continue to hold if we replace the partition $\mathcal{P}$ with any coarser partition (such as the trivial partition $[d]$). Thus, our results strictly generalize those about MGDA (*e.g.*, Fliege et al., 2019).

**Complexity.** We emphasize the inner loop of Algorithm 1 is executed in parallel, as they correspond to updating different blocks of $\mathbf{w}$ *simultaneously*. We have analyzed the complexity of the subroutine REFINED_PARTITION in the previous section, and we note that this one-time cost can often be avoided if one knows the domain application well, see our experiments in Section 7 for examples. Barring the one-time overhead in line 1, Algorithm 1 is theoretically cheaper than the vanilla MGDA, since the most costly step in line 5 (solving the dual subproblem) often has superlinear dependence on $m$ (the number of objectives) and hence dividing it into blocks of smaller sizes reduces the runtime.

## 7   EXPERIMENTS

In this section, we present experiments comparing RP-MGDA and MGDA across both synthetic examples and realistic benchmark datasets. These experiments explore varying variable dependency structures and convexity assumptions, demonstrating RP-MGDA's effectiveness and versatility. More experimental details and discussions can be found in Appendix B.

### 7.1   SYNTHETIC EXAMPLES

We examine two settings. First, quadratic MOO problems with randomly generated adjacency matrices, focusing on the behaviour of REFINED_PARTITION and the effectiveness of RP-MGDA. Second, in Appendix B.1, we consider a two-objective convex problem, where we visualize and compare solution fronts, observing how often MGDA converges to sub-optimal PS solutions.

**Randomly generated sparse adjacency matrix.** We randomly generate $m \times d$ adjacency matrices with varying degrees of sparsity to set up the function-variable dependency structures of quadratic MOO problems with $m$ objectives and $d$ input variables. The sparsity of function-variable dependencies is controlled by a 'density' parameter, which determines the percentage of non-zero entries in the adjacency matrix. We use our proposed REFINED_PARTITION procedure to automatically

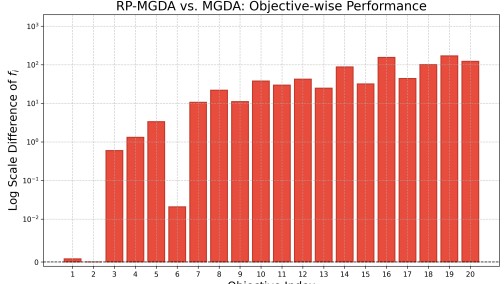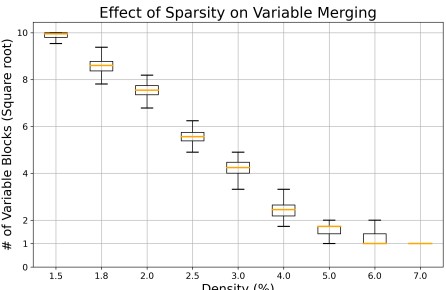

Figure 4: Experiments on randomly generated sparse adjacency matrices. **Left**: [$20 \times 100$ matrix]. The bar chart compares the solution from MGDA with that post-processed by RP-MGDA. Red bars indicate the magnitude by which RP-MGDA outperforms MGDA in objective value, while green bars (if any) indicate the opposite. **Right**: [$100 \times 100$ matrices]. The impact of adjacency matrix density on variable merging in REFINED_PARTITION, where $y$-axis shows the final number of variable blocks returned. A value of 1 indicates all variables are merged into a single block, and a value of 100 (represented as 10 on the square root scale) indicates that all variables are fully separated.

|       | $\mathbf{w}_0$ | $\mathbf{w}_1$ | $\mathbf{w}_2$ | $\mathbf{w}_3$ | $\mathbf{w}_4$ |
|-------|------|------|------|------|------|
| $f_1$ | *    | *    |      |      |      |
| $f_2$ | *    |      | *    |      |      |
| $f_3$ | *    |      |      | *    |      |
| $f_4$ | *    |      |      |      | *    |

|       | $\mathbf{w}_1$ | $\mathbf{w}_2$ | $\mathbf{w}_3$ |
|-------|------|------|------|
| $f_1$ | *    |      |      |
| $f_2$ | *    | *    |      |
| $f_3$ | *    | *    | *    |

|       | $\mathbf{w}_1$ | $\mathbf{w}_2$ | $\mathbf{w}_3$ |
|-------|------|------|------|
| $f_1$ | *    |      |      |
| $f_2$ | *    | *    |      |
| $f_3$ |      | *    | *    |
| $f_4$ |      |      | *    |

Figure 5: Different function-variable dependency structures for experiments on benchmark datasets. Left: Personalized federated learning; Middle: Hierarchical classification; Right: Multi-objective learning with partial information.

generate refined partitions, especially for larger adjacency matrices (*e.g.* $100 \times 100$) where manual partitioning becomes cumbersome. Details of the setup can be found in Appendix B.2.

**Results.** From Figure 4 (Left) we observe that by post-processing the outputs of MGDA, *i.e.*, using them to initialize RP-MGDA, we can improve the solutions to Pareto dominate the original ones. Conversely, we point out that MGDA cannot improve the stationary solutions of RP-MGDA, due to Theorem 1. In the demonstration with $100 \times 100$ matrix (Figure 4, Right), we note that the number of final blocks returned by REFINED_PARTITION decreases as the *density* increases, starting at $1.5\%$, where all variables are separated, and eventually merging all variables into a single block when the density reaches $7\%$. We also empirically verified Theorem 3 in Appendix B.2.

## 7.2 ML BENCHMARK DATASETS

We examine three problems with different function-variable dependency structures derived from real-world application scenarios. These examples also serve as a 'proof of concept' for applying refined partitioning to realistic ML problems. The structures and their corresponding refined partitions are illustrated in Figure 5.

### 7.2.1 PERSONALIZED FEDERATED LEARNING ('MTL' STRUCTURE)

**Setup.** We consider a personalized federated learning setting with $m = 4$ clients, each holding distinct non-i.i.d. data $D_i$ sampled from the training dataset (MNIST/CIFAR-10). The objectives $f_i$ are defined as empirical cross-entropy losses over the clients' data distributions. As in standard federated learning, we have a shared global model with learnable parameters $\mathbf{w}_0$. To handle data heterogeneity and enable personalization, instead of directly feeding inputs (e.g., images) into the global model $\mathbf{w}_0$, each client $i$ has a local personalized network, parameterized by $\mathbf{w}_i$, which extracts

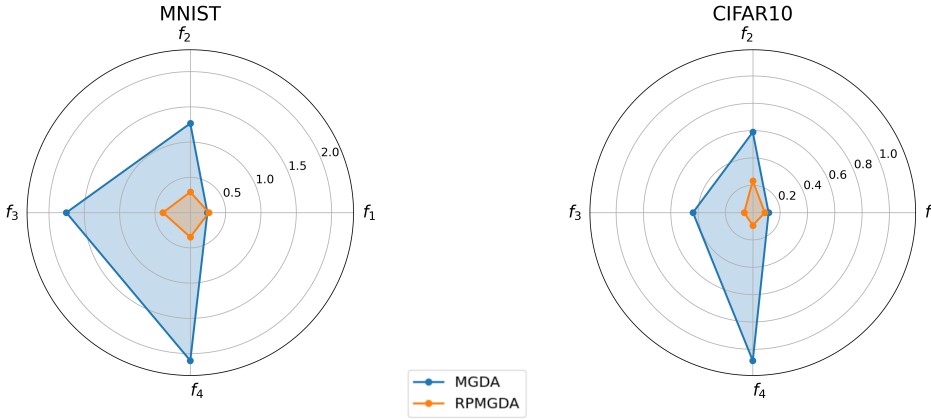

Figure 6: Radar graph of final cross-entropy losses $f_i$ in the personalized federated learning setting, using MGDA and RP-MGDA respectively. Lower is better.

client-specific representations. These representations are then forwarded to the global model for prediction (*e.g.*, Liang et al., 2020). Formally, $f_i(\mathbf{w}_0, \mathbf{w}_i) = \mathbb{E}_{\mathbf{x} \sim D_i} [\ell(\text{NN}(\text{NN}(\mathbf{x}; \mathbf{w}_i); \mathbf{w}_0))]$, thus having the dependency structure illustrated in Figure 5 (Left), which is also commonly seen in multi-task learning (MTL), see *e.g.*, Sener and Koltun (2018).

**Refined Partition.** For this 'MTL' dependency structure, the optimal refined partition is $\{\{\mathbf{w}_0\}, \{\mathbf{w}_1\}, \ldots, \{\mathbf{w}_m\}\}$. Consequently, at each iteration, RP-MGDA updates $\mathbf{w}_0$ by descending along the minimum-norm direction, while $\mathbf{w}_i$ (for $i \in [m]$) is updated with gradient descent *simultaneously*. This partitioning (a) simplifies the dual sub-problem by reducing its dimensionality and (b) allows for more flexible updates to the personalized networks.

**Results.** As shown in Figure 6 and Appendix B.3.2, we observe that RP-MGDA converges faster and reaches better solutions than MGDA, particularly for $f_2$, $f_3$ and $f_4$, while for $f_1$, the performances are similar. Further details are provided in Appendix B.3.

### 7.2.2 HIERARCHICAL CLASSIFICATION ('LADDER' STRUCTURE)

**Setup.** We consider a multi-objective hierarchical classification problem with $m = 3$ levels of objectives that predict image categories, from coarse to fine labels, on the CIFAR-10 dataset, using a network structure similar to Branch-CNN (Zhu and Bain, 2017). Here, $\mathbf{w}_1$, $\mathbf{w}_2$ and $\mathbf{w}_3$ represent the network weights of feature extraction layers $1, 2$ and $3$, respectively. The three objectives, $f_1$, $f_2$ and $f_3$, are all cross-entropy losses but defined for different prediction tasks. The first task is binary classification, categorizing images as 'living' or 'non-living,' using only the representations from feature extraction layer 1, and thus depending solely on $\mathbf{w}_1$.[4] The second task classifies images into four categories: 'ground vehicle,' 'non-ground vehicle,' 'land animal,' or 'non-land animal,' using representations from layers 1 and 2, and thus depending on $\mathbf{w}_1$ and $\mathbf{w}_2$. The third task classifies images into the original CIFAR-10 categories, using representations from all three layers, and thus depending on $\mathbf{w}_1$, $\mathbf{w}_2$, and $\mathbf{w}_3$. This hierarchical structure allows for progressively more complex predictions as we move from coarse to fine labels.

**Refined Partition.** The corresponding dependency structure in Figure 5 (Middle) features an optimal refined partition of $\{\{\mathbf{w}_1 \ \mathbf{w}_2\}, \{\mathbf{w}_3\}\}$, where RP-MGDA optimizes $\mathbf{w}_3$ using gradient descent and the block $\{\mathbf{w}_1 \ \mathbf{w}_2\}$ using MGDA. This partitioning should be advantageous for Objective 3, as it allows for more flexible updates to $\mathbf{w}_3$ compared to MGDA.

**Results.** Indeed, we observe that the training loss curves for Objectives 1 and 2 are nearly indistinguishable; however, RP-MGDA demonstrates faster and better convergence for Objective 3, as illustrated in Figure 7.

---

[4]After 'branching out,' an additional linear classification layer is used for the final prediction, which can either be fixed or updated separately. We omit these 'off-ramps' from the discussion, as they do not affect the dependency structure of the feature extraction layers, which is the focus here.

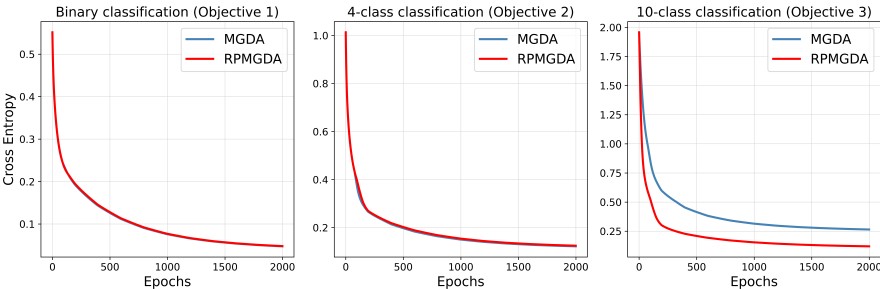

Figure 7: Loss curves of MGDA and RP-MGDA for the three objectives in hierarchical classification.

### 7.2.3 MULTI-OBJECTIVE LEARNING WITH PARTIAL INFORMATION ('CHAIN' STRUCTURE)

**Setup.** We simulate a scenario inspired by vertical federated learning (Liu et al., 2024) where each party holds different feature sets for the same instances, akin to multiple institutions possessing various attributes for the same users. In particular, we explore the setting where each party only has access to consecutive variables, *e.g.*, a company in a supply chain having only adjacent upstream and downstream data. In this context, clients with heterogeneous data collaborate to learn a stronger model without directly sharing sensitive user data. Specifically, we consider an $m = 4$-objective learning problem on the California housing dataset (Pace and Barry, 1997), where we employ a linear model to predict the median house value. The original 8 features are grouped into 3 sets[5], each associated with linear regression weights $\mathbf{w}_1, \mathbf{w}_2, \mathbf{w}_3$. The objectives $f_i$ represent the mean-squared error (MSE) for different parties, all trained on the same dataset but with each objective depending on different subsets of features, as illustrated in Figure 5 (Right).

**Refined Partition.** For this 'chain' dependency structure, the optimal refined partition is $\{\{\mathbf{w}_1\}, \{\mathbf{w}_2\}, \{\mathbf{w}_3\}\}$, allowing RP-MGDA to apply block-wise MGDA to $\mathbf{w}_1, \mathbf{w}_2$ and $\mathbf{w}_3$ separately. This results in simpler dual sub-problems and improved convergence.

**Results.** As shown in Figure 8, RP-MGDA demonstrates greater stability and lower average losses across the 20 trials. We also observe that the stationary solutions of MGDA can be improved by post-processing with RP-MGDA, in 18 out of 20 trials. Additionally, the conditions of Theorem 2 are met in this setup, ensuring that RP-MGDA achieves Pareto optimality.

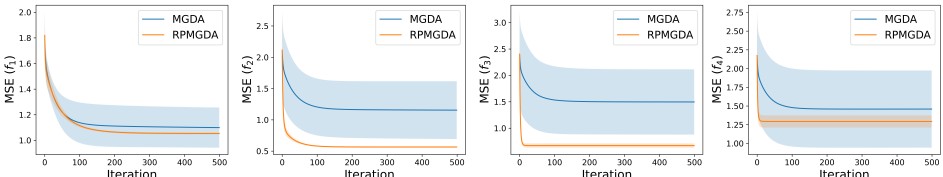

Figure 8: We calculate the mean and standard deviation of the 4 losses across 20 ramdom trials (different seeds) for both MGDA and RP-MGDA, and plot the means, with shaded regions representing the standard deviation.

## 8 CONCLUSION AND DISCUSSION

In this work, we introduced Refined Pareto Stationarity (RPS) with an efficient partitioning procedure (REFINED_PARTITION) to address limitations of Pareto stationarity in multi-objective optimization (MOO). We proposed RP-MGDA, a novel MOO algorithm based on refined partitioning that converges to RPS, supported by both theoretical and empirical justification. Future work may explore applying our idea to other MOO algorithms and demonstrate the superior performance on other real problems with (sparse) dependency structures.

---

[5]Details on feature grouping are provided in Appendix B.5.

## ACKNOWLEDGEMENTS

We thank the reviewers and the area chair for their constructive comments that have improved the presentation of this work. Zeou Hu would like to thank Dihong Jiang and Haoye Lu for helpful discussions. YY gratefully acknowledges CIFAR and NSERC for funding support.

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

## A   PROOFS AND PSEUDO-CODE OMITTED FROM THE MAIN TEXT

**Lemma 2.** *Any Pareto optimal solution is Pareto stationary w.r.t. any partition.*

*Proof.* Let $\mathbf{w}^*$ be Pareto optimal w.r.t. $\mathbf{f}$. Consider the function $\mathbf{g}(\mathbf{z}) := \mathbf{f}^{P_1}(\mathbf{z}, \mathbf{w}^*_{P_2}, \dots, \mathbf{w}^*_{P_k})$. Clearly, $\mathbf{w}^*_{P_1}$ is Pareto optimal w.r.t. $\mathbf{g}$. Applying Lemma 1 we know $\mathbf{0} \in \text{conv}\{\nabla \mathbf{g}(\mathbf{w}^*_{P_1})\} = \text{conv}\{\nabla_{\mathbf{w}_{P_1}} \mathbf{f}^{P_1}(\mathbf{w}^*)\}$. Iterating the argument proves (11). $\qquad\square$

**Theorem 1.** *Let $\mathcal{P} = \text{REFINED\_PARTITION}(A)$ where $A$ is defined in (12). Then, any $\mathbf{w}_*$ that is Pareto stationary w.r.t. $\mathcal{P}$, which we call refined Pareto stationary from now, is Pareto stationary.*

*Proof.* For each $j \in [k]$, let us define

$$F_j := \{i \in [m] : f_i \text{ is connected to } P_j \text{ in the final graph}\}, \tag{13}$$

namely, the set of functions that the variable set $P_j$ "sees." We claim that for all $j \neq l$,

$$|F_j \cap F_l| \leq 1. \tag{14}$$

Suppose not, let $a \neq b \in F_j \cap F_l$. Then, in our final graph, $f_a$ and $f_b$ are connected to both $P_j$ and $P_l$, creating a cycle of 4 and hence contradicting to the fact that $P_j$ and $P_l$ were not merged.

Since $\mathbf{w}_*$ is Pareto stationary w.r.t. $\mathcal{P}$, for each $j \in [k]$ there exists $\boldsymbol{\lambda}^j \in \Delta$ such that

$$\nabla_{\mathbf{w}_{P_j}} \mathbf{f}(\mathbf{w}_*) \cdot \boldsymbol{\lambda}^j = \mathbf{0}, \quad \text{where} \quad \lambda_i^j = 0 \text{ if } i \notin F_j. \tag{15}$$

Our goal is to construct $\boldsymbol{\lambda} \in \Delta$ (independent of the index $j$) such that $\nabla \mathbf{f}(\mathbf{w}_*) \cdot \boldsymbol{\lambda} = \mathbf{0}$, *i.e.*, $\mathbf{w}_*$ is Pareto stationary.

W.l.o.g. we make the following assumptions:

1. The graph is connected, as we can consider each connected component separately.

2. The variable sets are rearranged according to their distance to $P_1$ (ties broken arbitrarily).

3. (positivity) $\lambda_i^j = 0$ if and only if $i \notin F_j$; we will see how to reduce to this case later.

For each $j \in [k]$, consider any $n \geq j+1$ such that there exists a (unique) $i_n \in F_j \cap F_n$, and perform the rescaling

$$\boldsymbol{\lambda}^n \leftarrow \boldsymbol{\lambda}^n \cdot \frac{\lambda_{i_n}^j}{\lambda_{i_n}^n}, \tag{16}$$

where we note that $\lambda_{i_n}^n \neq 0$ due to the positivity assumption. Importantly, rescaling (by a positive number) does not affect any of the stationary conditions in (15), other than the normalization constraint $\mathbf{1}^\top \boldsymbol{\lambda}^j = 1$, which we will take care of at the end.

We claim that each $\boldsymbol{\lambda}^n$ is rescaled at most once. Suppose not so there exist $j_1 < j_2 < n$ and $i_{j_1} \in F_{j_1} \cap F_n$ and $i_{j_2} \in F_{j_2} \cap F_n$, *i.e.*, we have the edges shown on Figure 9 (left). Due to their appearance order, we must have

$$\text{dist}(P_{j_1}, P_1) \leq \text{dist}(P_{j_2}, P_1) \leq \text{dist}(P_n, P_1). \tag{17}$$

Noting the edges from $i_{j_1}$ to $P_{j_1}$ and $P_n$ and from $i_{j_2}$ to $P_{j_2}$ and $P_n$, we must have

$$\text{dist}(P_{j_1}, P_1) = \text{dist}(P_{j_2}, P_1) < \text{dist}(P_n, P_1), \tag{18}$$

for otherwise there will be a cycle (due to a back edge, see Cormen et al., 2009). We can now trace back the ancestors of $P_{j_1}$ and $P_{j_2}$ until we finally arrive at the same variable set $P$ (since only $P_1$ has 0 distance to $P_1$) that intersects both $P'_{j_1}$ and $P'_{j_2}$ (here prime denotes the final ancestor pair). But this creates a cycle (see Figure 9, right), contradiction.

Lastly, we take component-wise maximum to construct

$$\boldsymbol{\lambda} = \vee_{j=1}^k \boldsymbol{\lambda}^j. \tag{19}$$

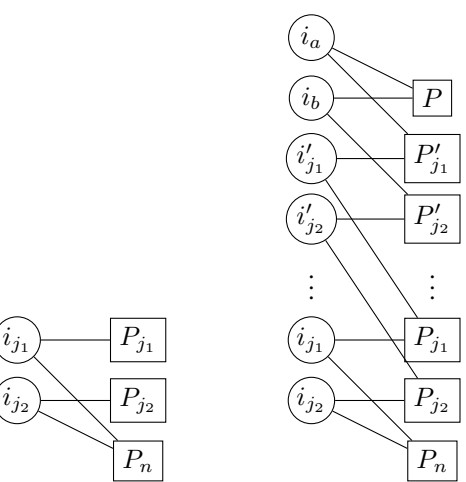

Figure 9: Illustration of the claim that each $\boldsymbol{\lambda}^n$ is rescaled at most once.

Clearly, $\boldsymbol{\lambda} \neq \mathbf{0}$ so we may normalize it so that $\boldsymbol{\lambda} \in \Delta$. Since each $\boldsymbol{\lambda}^j$ is re-scaled at most once (and nonzero entries of different $\boldsymbol{\lambda}^j$ and $\boldsymbol{\lambda}^n$ intersect at most once, see (14)), for any $j$ and $\lambda_i^j \neq 0$, we have

$$\lambda_i = \lambda_i^j. \tag{20}$$

Since (15) holds for any $j$ (other than the normalization constraint), it follows that $\nabla \mathbf{f}(\mathbf{w}_*) \cdot \boldsymbol{\lambda} = \mathbf{0}$.

We are left to justify the positivity assumption. Consider the "adjacency" matrix

$$\Lambda = \begin{bmatrix} \boldsymbol{\lambda}^1 & \boldsymbol{\lambda}^2 & \cdots & \boldsymbol{\lambda}^k \end{bmatrix}, \tag{21}$$

where we put an edge between $f_i$ and $P_j$ if the corresponding $\lambda_i^j$ value is nonzero. Suppose $\lambda_i^j = 0$ for some $i \in F_j$. Then, for all variable subsets $P_l$, other than $P_j$, that can be reached by $f_i$, we set their $\boldsymbol{\lambda}^l = \mathbf{0}$ (namely entire columns in $\Lambda$) and $F_l = \emptyset$. Note that we do not change $\boldsymbol{\lambda}^j$ (for if we can reach $P_j$ from $i$ but not through the edge $\{i, P_j\}$, we will have a cycle) so $\Lambda$ remains nonzero after our operation. Moreover, the stationary conditions (15), other than the normalization constraint, are maintained. Thus, we may repeat this process until the positivity assumption holds.

The proof is now complete. □

**Remark 1.** *We point out that* REFINED\_PARTITION*(A) returns a unique solution (irrespective of the order of the cycles we contract). Indeed, when we merge the variables and contract the graph, any existing cycle (other than the one we are contracting) will remain as a cycle. Another way to see this is to think of contracting a cycle as replacing it with a complete bipartite subgraph.*

**Remark 2.** *We note that Theorem 1 continues to hold if we replace $\mathcal{P}$ by any of its coarser partition $\mathcal{Q}$ (i.e., for any $P \in \mathcal{P}$ there exists $Q \in \mathcal{Q}$ such that $P \subseteq Q$). In particular, setting $\mathcal{Q} = [d]$ we recover Pareto stationarity in Definition 3.*

We will need the following elementary result:

**Lemma 3.** *If $g(\mathbf{w}, \mathbf{u})$ and $h(\mathbf{w}, \mathbf{v})$ are both strictly convex in $(\mathbf{w}, \mathbf{u})$ and $(\mathbf{w}, \mathbf{v})$, respectively, then $f(\mathbf{w}, \mathbf{u}, \mathbf{v}) := g(\mathbf{w}, \mathbf{u}) + h(\mathbf{w}, \mathbf{v})$ is strictly convex in $(\mathbf{w}, \mathbf{u}, \mathbf{v})$.*

*Proof.* Indeed, consider $(\mathbf{w}_0, \mathbf{u}_0, \mathbf{v}_0) \neq (\mathbf{w}_1, \mathbf{u}_1, \mathbf{v}_1)$. For any $\lambda \in (0, 1)$, denote $\mathbf{w}_\lambda := (1 - \lambda)\mathbf{w}_0 + \lambda\mathbf{w}_1$ and similarly for $\mathbf{u}_\lambda$ and $\mathbf{v}_\lambda$, we have

$$g(\mathbf{w}_\lambda, \mathbf{u}_\lambda) \leq (1 - \lambda)g(\mathbf{w}_0, \mathbf{u}_0) + \lambda g(\mathbf{w}_1, \mathbf{u}_1) \tag{22}$$
$$h(\mathbf{w}_\lambda, \mathbf{v}_\lambda) \leq (1 - \lambda)h(\mathbf{w}_0, \mathbf{v}_0) + \lambda h(\mathbf{w}_1, \mathbf{v}_1), \tag{23}$$

where at least one of the inequalities is strict. Thus, $f(\mathbf{w}_\lambda, \mathbf{u}_\lambda, \mathbf{v}_\lambda) < (1 - \lambda)f(\mathbf{w}_0, \mathbf{u}_0, \mathbf{v}_0) + \lambda f(\mathbf{w}_1, \mathbf{u}_1, \mathbf{v}_1)$. □

**Theorem 2.** *Let $\mathcal{P} = \{P_1, \ldots, P_k\}$ be returned by* REFINED_PARTITION. *Suppose each $f_i$ is strictly convex in $\mathbf{w}_{I_i}$ where $I_i \subseteq \mathcal{P}$ is the set of blocks of variables that $f_i$ depends on. Then, any refined Pareto stationary solution (w.r.t. $\mathcal{P}$) is Pareto optimal.*

*Proof.* Let $\mathbf{w}^*$ be an arbitrary refined Pareto stationary solution w.r.t. $\mathcal{P}$. For the sake of contradiction, consider any Pareto optimal solution $\mathbf{z}$ that dominates $\mathbf{w}^*$, *i.e.*, $\mathbf{f}(\mathbf{z}) \lneq \mathbf{f}(\mathbf{w}^*)$.

Applying Theorem 1 we know $\mathbf{w}^*$ is Pareto stationary, *i.e.*, there exists some $\boldsymbol{\lambda} \in \Delta$ such that

$$\sum_i \lambda_i \nabla f_i(\mathbf{w}^*) = 0. \tag{24}$$

Restricting to the subset $F^+ := \{i : \lambda_i \neq 0\} \neq \emptyset$, we know from Lemma 3 that the function $h(\mathbf{w}) := \sum_{i \in F^+} \lambda_i f_i(\mathbf{w})$ is strictly convex in $\mathbf{w}_{I_h}$ that it depends on (where $I_h \subseteq \mathcal{P}$). Since $h(\mathbf{z}) \leq h(\mathbf{w}^*)$, we necessarily have

$$\mathbf{z}_{\cup I_h} = \mathbf{w}^*_{\cup I_h}. \tag{25}$$

We are now ready to reduce. We fix $\mathbf{w}^*_{\cup I_h}$ and remove all blocks of variables in $I_h$ (and the corresponding edges in our bipartite graph). Clearly, the bipartite graph remains acyclic and $\mathbf{w}^*_{\mathcal{P} \setminus I_h}$ remains to be refined Pareto stationary w.r.t. the reduced problem. Iterating the argument in the previous paragraph, we deduce that $\mathbf{z}$ agrees with $\mathbf{w}^*$ in at least one more block of variables. Since there are only $k$ blocks in total, after at most $k$ iterations we must have $\mathbf{z} = \mathbf{w}^*$, which contradicts to the assumption $\mathbf{f}(\mathbf{z}) \lneq \mathbf{f}(\mathbf{w}^*)$. Thus, $\mathbf{w}^*$ is Pareto optimal and the proof is complete. $\square$

**Remark 3.** *Since Theorem 1 also holds for any partition $\mathcal{Q}$ that is coarser than $\mathcal{P}$, the same is true for Theorem 2. In particular, choosing $\mathcal{Q} = [m]$, refined Pareto stationarity reduces to Pareto stationarity and Theorem 2 reduces to the well-known result in* MOO *(see Lemma 1).*

**Example 3** (Personalized federated learning). *Let us give an example where Theorem 2 strictly improves Lemma 1. Consider the personalized federated learning setting with $f_i(\mathbf{w}_0, \mathbf{w}_i), i \in [m]$. It is clear that none of the functions $f_i$ is strictly convex w.r.t. $(\mathbf{w}_0, \mathbf{w}_1, \ldots, \mathbf{w}_m)$. Thus, Lemma 1 is not applicable and not every Pareto stationary solution is Pareto optimal.*

*On the other hand,* REFINED_PARTITION *returns $\mathcal{P} = \{\{\mathbf{w}_0\}, \{\mathbf{w}_1\}, \ldots, \{\mathbf{w}_m\}\}$ and it is possible for each function $f_i$ to be strictly convex w.r.t. the (blocks of) variables $(\mathbf{w}_0, \mathbf{w}_i)$ that it depends on. (This can also be achieved by regularizing each function $f_i$.) Therefore, Theorem 2 is applicable. In fact, we can apply Theorem 2 to both our Example 1 and Example 2 in the main text and conclude that the refined Pareto stationary solution indeed coincides with the unique Pareto optimal solution.*

*To be more concrete, consider*

$$f_1(w_0, w_1, w_2, w_3) = w_0^2 + w_1^2 \tag{26}$$

$$f_2(w_0, w_1, w_2, w_3) = (w_0 - 1)^2 + w_2^2 \tag{27}$$

$$f_3(w_0, w_1, w_2, w_3) = (w_0 - 2)^2 + w_3^2. \tag{28}$$

*We run simulations with step size $\eta = 0.01$ and 1000 iterations. With initialization $\mathbf{w}^1 = (1.5, 1, 0.1, 0.1)$, MGDA converges to $\mathbf{w}^{1000} = (1.5, 1, 0, 0)$ with function values $\mathbf{f}(\mathbf{w}^{1000}) = (\frac{13}{4}, \frac{1}{4}, \frac{1}{4})$, which is not Pareto optimal since it is dominated by $\mathbf{f}^* = (\frac{9}{4}, \frac{1}{4}, \frac{1}{4})$, achieved at $\mathbf{w}^* = (1.5, 0, 0, 0)$. In contrast, under the same setting, RP-MGDA converges to the Pareto optimal solution $\mathbf{w}^* = (1.5, 0, 0, 0)$.*

*From Theorem 2, we also see how MGDA fails to reach the Pareto optimal solution in this example. When MGDA converges to $\mathbf{w}^{1000} = (1.5, 1, 0, 0)$, the corresponding $\lambda_1 = 0$ while $\nabla_{w_1} f_1(w_0^*, w_1^*) \neq 0$, which means MGDA stops even though there is a descent direction along $w_1$.*

**Theorem 3.** *Suppose the functions $f_i$ are bounded from below and Lipschitz smooth with constant $L_i$. Set $\eta_j \equiv \eta \leq \min_i \frac{1}{L_i}$. Then, $\min_{1 \leq t \leq T} \|\mathbf{d}^t\| \leq O(\frac{1}{\sqrt{T}})$.*

*Proof.* Denote

$$I_i := \{j : f_i \text{ depends on } P_j\} \tag{29}$$

and apply smoothness to upper bound the $i$-th objective as (for step size $\eta \leq \frac{1}{L_i}$, $L_i$ being the Lipschitz constant of $\nabla f_i$):

$$f_i(\mathbf{w}^{t+1}) \leq f_i(\mathbf{w}^t) + \nabla f_i(\mathbf{w}^t) \cdot (\mathbf{w}^{t+1} - \mathbf{w}^t) + \frac{1}{2\eta}\|\mathbf{w}^{t+1} - \mathbf{w}^t\|_{I_i}^2 \tag{30}$$

$$= f_i(\mathbf{w}^t) + \sum_{j \in I_i} \eta \cdot \left[ \nabla_{P_j} f_i(\mathbf{w}^t) \cdot \mathbf{d}_{P_j}^t + \tfrac{1}{2}\|\mathbf{d}_{P_j}^t\|_2^2 \right] \tag{31}$$

$$\leq f_i(\mathbf{w}^t) - \sum_{j \in I_i} \tfrac{\eta}{2} \cdot \|\mathbf{d}_{P_j}^t\|_2^2, \tag{32}$$

where the last inequality follows from the definition of $\mathbf{d}_{P_j}^t$:

$$\mathbf{d}_{P_j}^t = \operatorname*{argmin}_{\mathbf{d}_{P_j}} \max_{i \in F_j} \left[ \nabla_{P_j} f_i(\mathbf{w}^t) \cdot \mathbf{d}_{P_j} + \tfrac{1}{2}\|\mathbf{d}_{P_j}\|_2^2 \right]. \tag{33}$$

Indeed, we note that the objective above is 1-strongly convex in $\mathbf{d}_{P_j}$, so setting $\mathbf{d}_{P_j} = \mathbf{0}$ and $\mathbf{d}_{P_j} = \mathbf{d}_{P_j}^t$ we obtain from *e.g.* Nesterov (2018, Corollary 3.2.3, p. 210):

$$0 \geq \max_{i \in F_j} \left[ \nabla_{P_j} f_i(\mathbf{w}^t) \cdot \mathbf{d}_{P_j}^t + \tfrac{1}{2}\|\mathbf{d}_{P_j}^t\|_2^2 \right] + \tfrac{1}{2}\|\mathbf{0} - \mathbf{d}_{P_j}^t\|_2^2. \tag{34}$$

Rearranging and summing over $t$ we obtain:

$$\sum_{t=1}^{T} \sum_{j \in I_i} \|\mathbf{d}_{P_j}^t\|_2^2 \leq \tfrac{2}{\eta}[f_i(\mathbf{w}^1) - f_i(\mathbf{w}^{T+1})]. \tag{35}$$

Dividing both sides by $T$ we obtain:

$$\min_{1 \leq t \leq T} \sum_{j \in I_i} \|\mathbf{d}_{P_j}^t\|_2^2 \leq \frac{1}{T} \sum_{t=1}^{T} \sum_{j \in I_i} \|\mathbf{d}_{P_j}^t\|_2^2 \leq \tfrac{2}{\eta T}[f_i(\mathbf{w}^1) - f_i(\mathbf{w}^{T+1})]. \tag{36}$$

Finally, we sum over any subset $F$ (*e.g.*, $F = [m]$) of functions that together depend on all variables:

$$\frac{2}{\eta} \sum_{i \in F}[f_i(\mathbf{w}^1) - f_i(\mathbf{w}^{T+1})] \geq \sum_{t=1}^{T} \sum_{i \in F} \sum_{j \in I_i} \|\mathbf{d}_{P_j}^t\|_2^2 \geq \sum_{t=1}^{T} \sum_{j=1}^{k} \|\mathbf{d}_{P_j}^t\|_2^2 = \sum_{t=1}^{T} \|\mathbf{d}^t\|_2^2. \tag{37}$$

Dividing both sides by $T$ we obtain:

$$\frac{2}{\eta T} \sum_{i \in F}[f_i(\mathbf{w}^1) - f_i(\mathbf{w}^{T+1})] \geq \frac{1}{T} \sum_{t=1}^{T} \|\mathbf{d}^t\|_2^2 \geq \min_t \|\mathbf{d}^t\|_2^2. \tag{38}$$

Since all functions are assumed to be bounded from below, the claim is proved. $\qquad\square$

### A.1 Pseudo-code for Refined_Partition

In the pseudocode below, a cycle $C$ is a collection of column indices (i.e. variable indices) of the adjacency matrix $A \in \mathbb{R}^{m \times d}$. We can use depth-first search (DFS) or networkx for `detect_cycle`. The loop can continue for at most $d$ iterations, with each iteration costing $O(m + d)$.

---

**Algorithm 2: Refined_Partition**

---

**Input:** Adjacency matrix $A$ representing the function-variable dependencies (bipartite graph)
**Output:** Refined partition $\mathcal{P}$ of variables

1 Initialize $\mathcal{P} \leftarrow \{\{w_1\}, \{w_2\}, \ldots, \{w_d\}\}$      // each variable in its own group
2 **while** *there exists a cycle* **do**
3      $C \leftarrow$ detect_cycle$(A)$      // e.g., use networkx package or DFS
4      $P \leftarrow \cup_{j \in C} \mathcal{P}_j$      // merge variables in the cycle
5      $\mathbf{a} \leftarrow \bigvee_{j \in C} A_{:j}$      // update the edges
6      $A_{:C} \leftarrow [], \ A \leftarrow [A, \mathbf{a}]$      // contract the graph
7      $\mathcal{P}_C \leftarrow [], \ \mathcal{P} \leftarrow \mathcal{P} \cup \{P\}$      // update the partition
8 **return** $\mathcal{P}$

---

## B EXPERIMENT DETAILS

### B.1 SYNTHETIC EXAMPLES

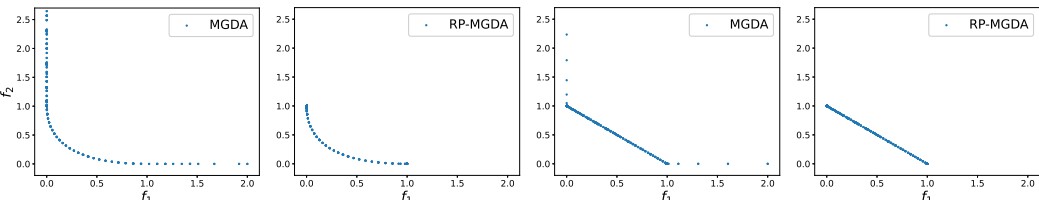

Figure 10: Comparisons of stationary solution fronts generated by MGDA and RPMGDA. **(1)** and **(2)** for the quadratic example, **(3)** and **(4)** for the non-strictly convex example.

**Two-objective toy examples.** We examine two examples with the same underlying function-variable dependency structure but different convexity conditions. The quadratic one is given by: $f_1(\mathbf{w}) = w_1^2 + w_2^2$, $f_2(\mathbf{w}) = (w_2 - 1)^2 + (w_3 - 1)^2$. The non-strictly convex one is: $f_1(\mathbf{w}) = w_1^2 + |w_2|$, $f_2(\mathbf{w}) = |w_2 - 1| + (w_3 - 1)^2$. Initialized with a grid search over $[-1, 1]^3$, we plot the solution fronts in terms of function values $[f_1(\mathbf{w}^*), f_2(\mathbf{w}^*)]$ for MGDA and RP-MGDA in Figure 10.

We observe that RP-MGDA always reaches Pareto optimal solutions regardless of initialization, while MGDA may converge to sub-optimal PS solutions (*i.e.*, those with $f_1 = 0, f_2 > 1$ and $f_1 > 1, f_2 = 0$).[6] Moreover, by post-processing the sub-optimal solutions of MGDA with RP-MGDA, we can refine these solutions and bring them to the Pareto front.

**Adding regularization to induce strict convexity.** We have discussed in Section 4 the importance of *joint* strict convexity for the performance of Pareto stationary solutions, particularly as highlighted in the second part of Lemma 1. We note that the functions in the two-objective toy example setting in Appendix B.1 are not strictly convex w.r.t. $\mathbf{w}$, though they are strictly convex with respect to their dependent variables (in the quadratic case). One natural attempt is to add regularization to the objectives to induce strict (indeed strong) convexity. For the two-objective toy example setting in Appendix B.1, we regularize the objectives with L2-norm on $\mathbf{w} = (w_1, w_2, w_3)$ with different magnitudes of regularization $\gamma$, and examine the new stationary solution fronts generated by MGDA. Specifically, we consider $f_1^\gamma(\mathbf{w}) := w_1^2 + w_2^2 + \gamma\|\mathbf{w}\|_2^2$, $f_2^\gamma(\mathbf{w}) = (w_2 - 1)^2 + (w_3 - 1)^2 + \gamma\|\mathbf{w}\|_2^2$.

We observe in Figure 11 that bigger regularization yields solutions that are further away from Pareto front; while smaller regularization gradually recover the previous (undesired) stationary solution set of MGDA (as in Figure 10), thus still having those inferior weakly Pareto solutions (*i.e.*, those on the tails). To summarize, adding a regularization term, whether large or small, to induce strong convexity, does *not* alleviate the issues with MGDA, further highlighting the effectiveness of our refined partitioning for MGDA.

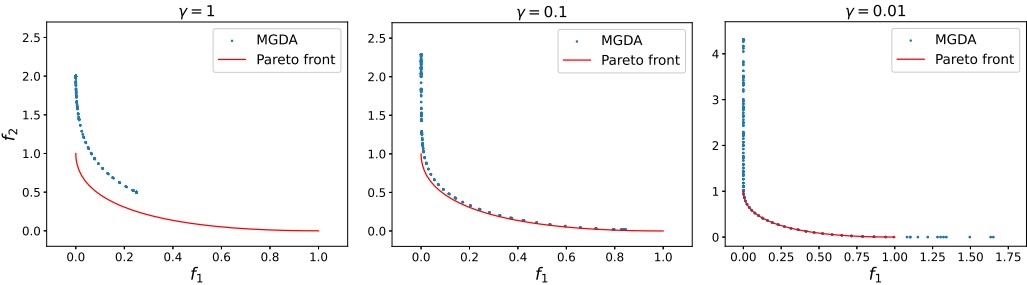

Figure 11: Stationary solution fronts of MGDA with regularized objectives, compared to the true Pareto front (red). From left to right: regularization magnitude $\gamma = 1, 0.1$ and $0.01$.

---

[6]For MGDA, 372/512 initializations converge to sub-optimal PS solutions in the strictly convex example.

## B.2 RANDOMLY GENERATED SPARSE ADJACENCY MATRIX

**Setup.** The problem is formulated by first generating a random sparse adjacency matrix of $m \times d$, with density $\rho$. In detail, we do this by randomly assigning $md\rho$ (rounded to integer) non-zero entries to a zero $m \times d$ matrix. Afterwards, we define our objectives $f_i = \sum_{w_j \in v(f_i)} (w_j - i)^2, i \in [m]$, where $v(f_i)$ is the set of variables that $f_i$ depends on.

**More results.** We compare the solution of MGDA and RP-MGDA with the same random initialization $\mathbf{w}_0$ (see Figure 12 Left), and notice that RP-MGDA outperforms MGDA in most objectives. Although RP-MGDA is theoretically superior, their different update rules can lead to incomparable solutions, even when the solution of RP-MGDA is Pareto optimal and MGDA is not.

We track the norm $\|\mathbf{d}^t\|$ of RP-MGDA and MGDA during optimization, and observe that both converge to zero. Note that the gradient norm $\|\mathbf{d}^t\|$ of RP-MGDA is much larger than MGDA in the beginning, which is not surprising since MGDA finds the overall min-norm element. Moreover, a descent direction $\mathbf{d}^t$ with a larger norm generally indicates a greater improvement for each update.

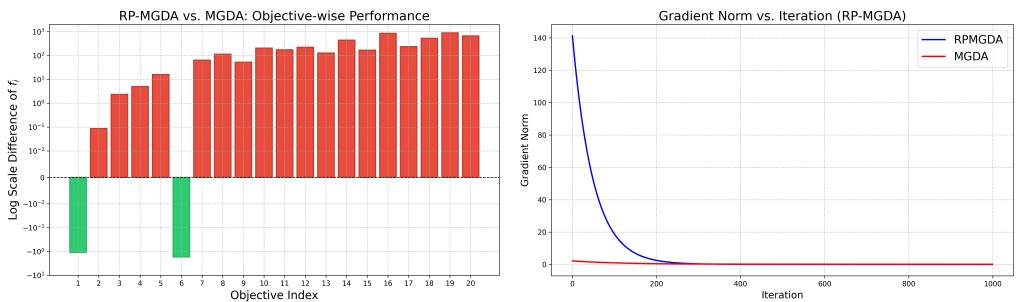

Figure 12: Left: compare MGDA and RP-MGDA solution, with same random initialization $\mathbf{w}_0$. Right: The norm of $\mathbf{d}^t$ over time converges to zero, empirically verifying Theorem 3.

We also present a $5 \times 5$ toy example below, which maybe useful for readers.

**Toy $5 \times 5$ dependency matrix.** Here we consider the quadratic objectives: $f_i = \sum_{w_j \in v(f_i)} (w_j - i)^2, i \in [5]$, where $v(f_i)$ is the set of variables that $f_i$ depends on. The variable dependency structure is given in Figure 32, Appendix B.7. The refined partition for this problem is $\{w_1, w_2, [w_3 \ w_4 \ w_5]\}$, where RP-MGDA is performing (8) w.r.t. $\mathbf{w}' = (w_3, w_4, w_5)$ and gradient descent w.r.t $w_1$ and $w_2$. We observe from Figure 13 that the solutions of RP-MGDA dominate those of MGDA, although all being Pareto stationary (with corresponding convex combination coefficients plotted).

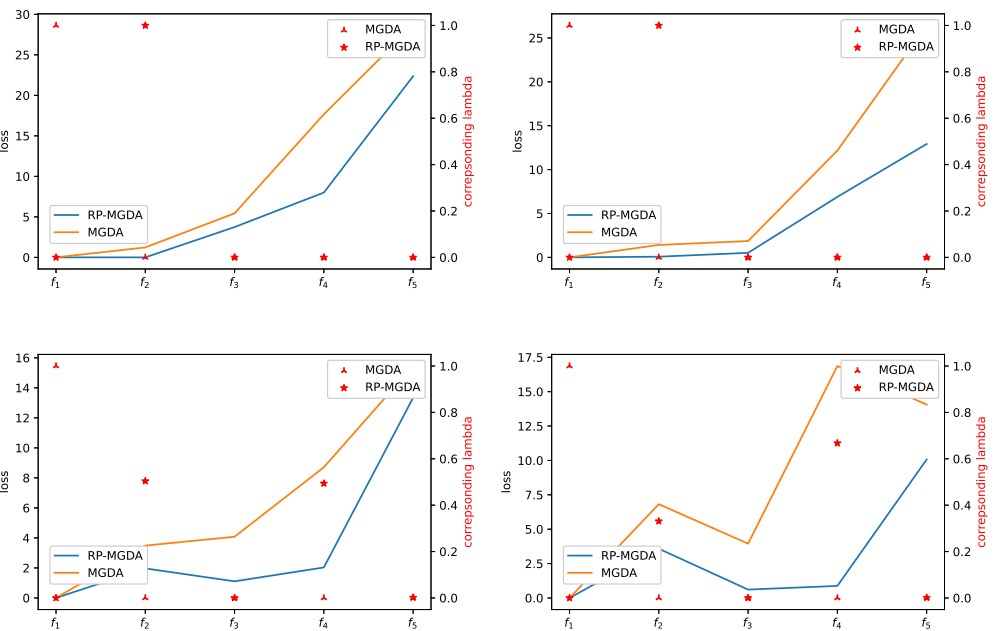

Figure 13: Comparison of MGDA and RP-MGDA solutions, with different initializations. **(i)** Colored lines connect the final objective values of each $f_i$, where blue represents RP-MGDA, and orange represents MGDA. **(ii)** Red markers (with values on the right $y$-axis) are the corresponding dual coefficients $\lambda_i$ of the solutions, showing the differences of solutions from a dual perspective.

### B.3  PERSONALIZED FEDERATED LEARNING (THE 'MTL' STRUCTURE)

In this section, we present detailed experiment setup and more results regarding the personalized federated learning (PFL) experiments.

#### B.3.1  DETAILED EXPERIMENT SETUP

**Data.** For the PFL setting, we conducted experiments on both MNIST and CIFAR-10 dataset. In order to create a non-i.i.d. dataset, we follow a similar sampling procedure as in McMahan et al. (2017): First we sort all data points according to their classes. Then, they are split consecutively into shards (300 shards for MNIST, 250 shards for CIFAR-10), with 200 images per shard, each shard contains images from only one class. Each client is randomly assigned 10 different shards, totaling 2000 instances per client. The data distribution for each client varies, with each having access to different subsets of class labels. For example, client 1's data includes class labels ['0' '2' '4' '6' '7' '8' '9'], while lacking data from class labels '1','3' and '5'.

**Model.** For MNIST, we use Multi-Layer Perceptrons (MLP) for both the lower personalized networks and the top global network. For CIFAR-10, the lower personalized networks are Convolutional Neural Networks (CNNs), while the top global network remains an MLP. Input data first passes through the lower personalized networks, and the resulting representations are then forwarded to the top global model for predictions. See Table 1 and 2 for configurations.

**Misc.** We adopt a warmstart procedure by first running RP-MGDA for 50 epochs on a given random initialization, using the resulting weights as the starting point for both MGDA and RP-MGDA training. This approach helps improve stability in some cases while still ensuring a fair comparison. We also employ a periodic exponential decay learning rate scheme for the CIFAR-10 experiments.

Table 1: Network architecture for PFL MNIST experiments.

| Network | Input Dim | Hidden Dim | Output Dim |
|---|---|---|---|
| Lower Personalized MLP | img_size | 256 | 128 |
| Top Global MLP | 128 | 128 | 10 |

Table 2: Network architecture for PFL CIFAR-10 experiments.

| Layer Type | Configuration | Activation |
|---|---|---|
| **Lower Personalized CNN Network** | | |
| Conv2D | $3 \times 32$, kernel size 5 | ReLU |
| BatchNorm2D | 32 channels | - |
| MaxPool2D | kernel size 2, stride 2 | - |
| Conv2D | $32 \times 32$, kernel size 5 | ReLU |
| BatchNorm2D | 32 channels | - |
| MaxPool2D | kernel size 2, stride 2 | - |
| Fully Connected | $32 \times 5 \times 5 \rightarrow 384$ | ReLU |
| **Top Global MLP Network** | | |
| Fully Connected | $384 \rightarrow 192$ | ReLU |
| Fully Connected | $192 \rightarrow 10$ | - |

### B.3.2 MORE RESULTS

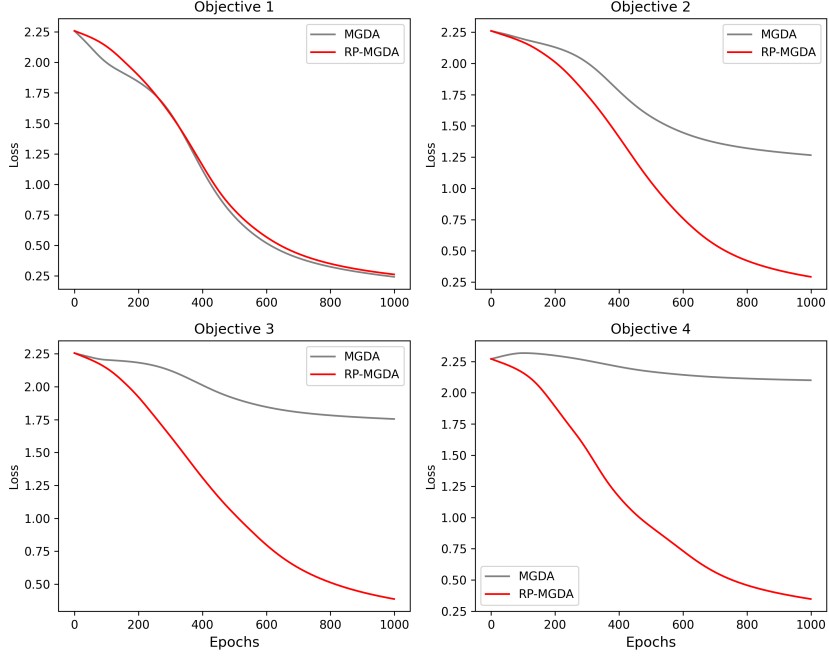

Figure 14: Personalized federated learning problem on MNIST. Comparing 4 objectives using RP-MGDA and MGDA. Random seed 1.

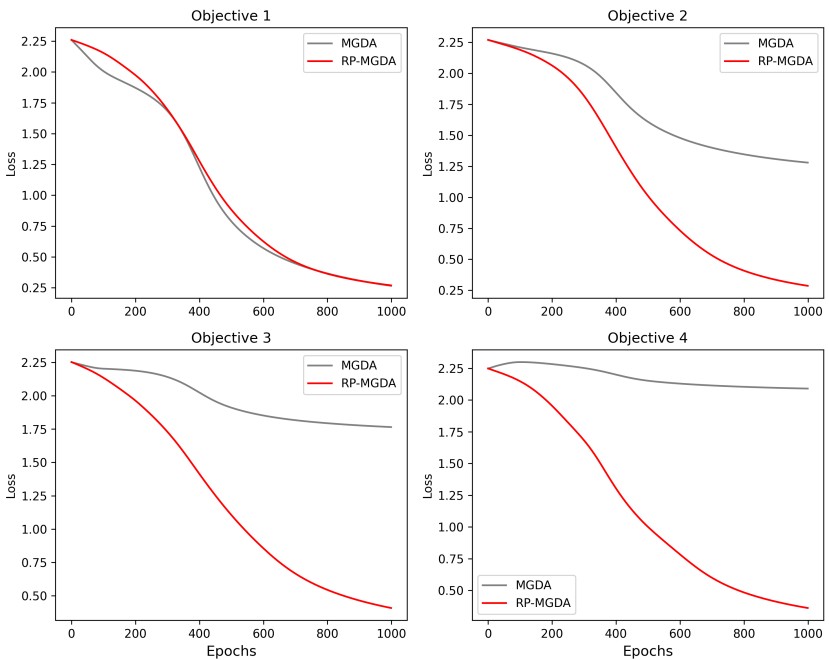

Figure 15: Personalized federated learning problem on MNIST. Comparing 4 objectives using RP-MGDA and MGDA. Random seed 2.

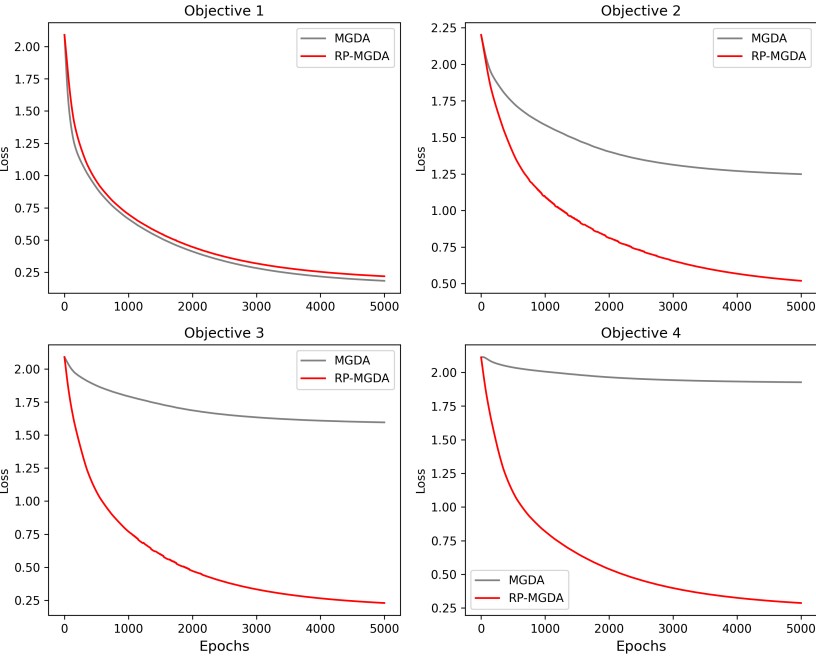

Figure 16: Personalized federated learning problem on CIFAR10. Comparing 4 objectives using RP-MGDA and MGDA. First random seed.

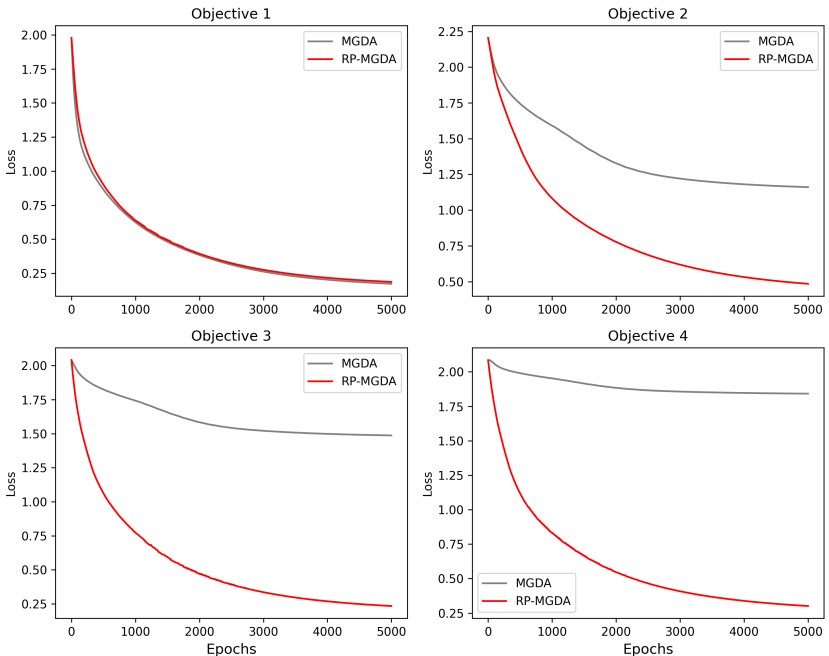

Figure 17: Personalized federated learning problem on CIFAR10. Comparing 4 objectives using RP-MGDA and MGDA. Random seed 2.

### B.3.3 ELU ACTIVATION

The purpose of these experiments is to verify whether replacing ReLU with differentiable ELU activations yields similar results and conclusions. We observed that the performance of both RP-MGDA and MGDA remains consistent with their performance when using ReLU, with RP-MGDA still outperforming MGDA.

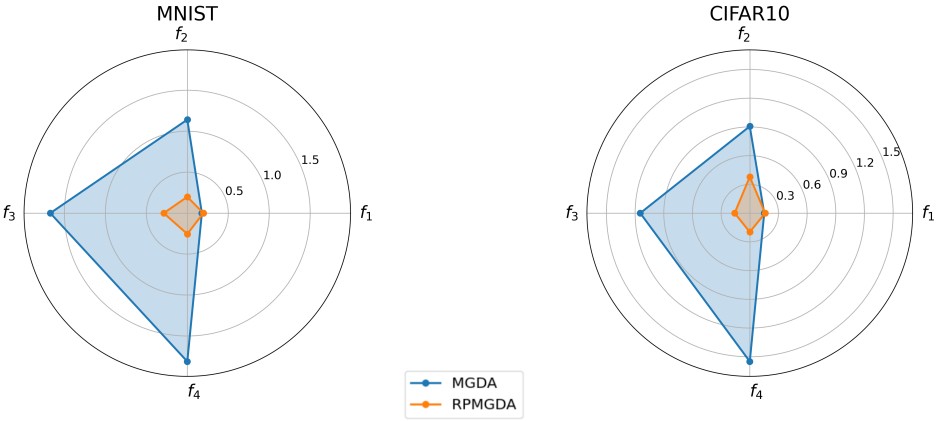

Figure 18: With ELU activation. Radar graph of final cross-entropy losses $f_i$ in the personalized federated learning setting, using MGDA and RP-MGDA respectively. Lower is better.

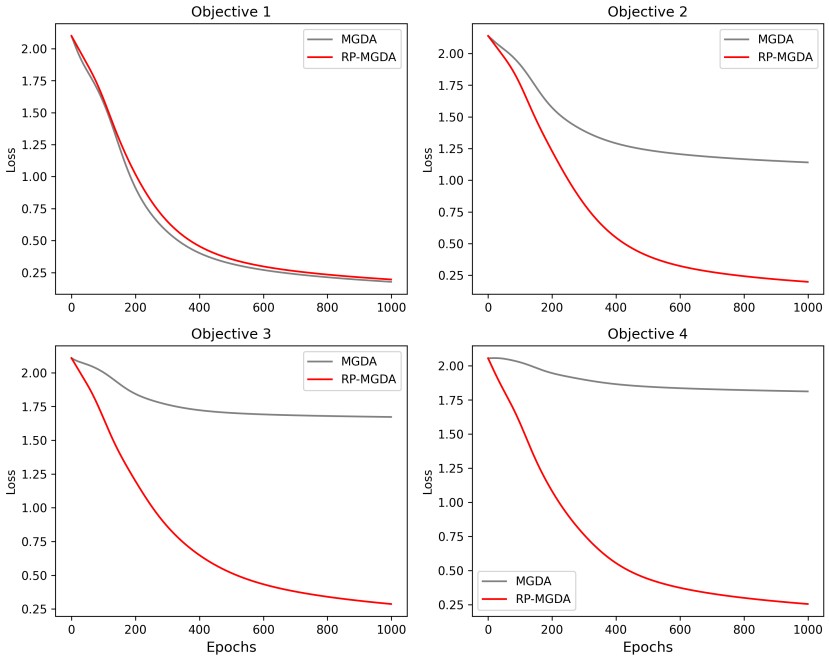

Figure 19: Personalized federated learning problem on MNIST. Comparing 4 objectives using RP-MGDA and MGDA. ELU activation.

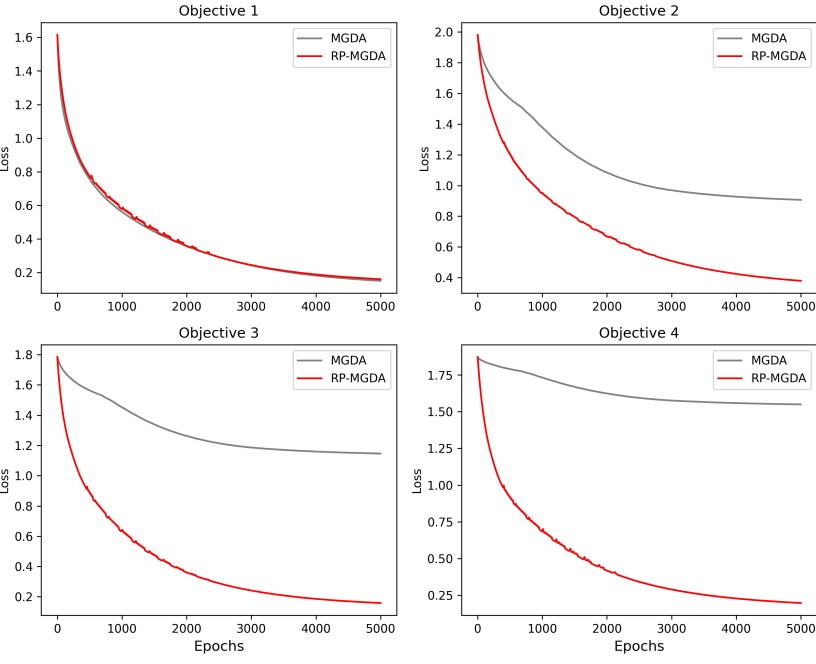

Figure 20: Personalized federated learning problem on CIFAR-10. Comparing 4 objectives using RP-MGDA and MGDA. ELU activation.

### B.3.4 STOCHASTIC GRADIENTS

We evaluate both RP-MGDA and MGDA in the stochastic setting, where only stochastic gradients are available due to mini-batch training. The experiments are run for 500 epochs, with each epoch corresponding to a full pass through the dataset. Mini-batch size= 200. For each epoch, we report the loss at the start of the epoch. For radar graph, we report the loss averaged across mini-batches of the last epoch.

We observe that RP-MGDA continues to outperform MGDA in performance. However, neither algorithm guarantees descent in every iteration, and the performance of both methods drops slightly, potentially due to bias in the descent direction.

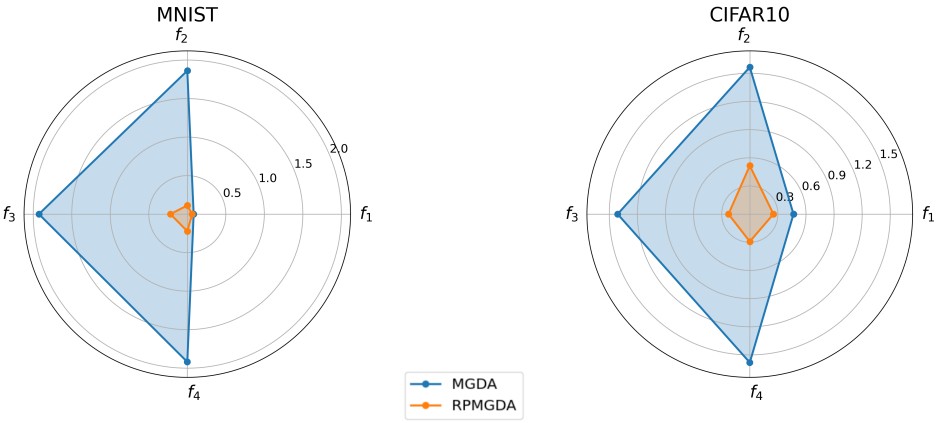

Figure 21: Stochastic gradients. Radar graph of final cross-entropy losses $f_i$ in the personalized federated learning setting, using MGDA and RP-MGDA respectively. Lower is better.

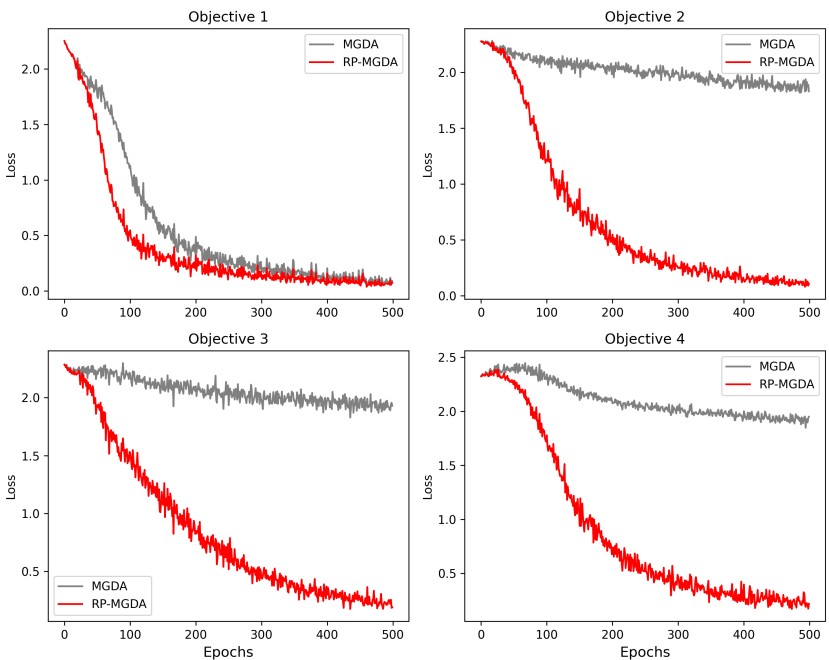

Figure 22: Personalized federated learning problem on MNIST. Comparing 4 objectives using stochastic RP-MGDA and stochastic MGDA. Report loss at the start of each epoch.

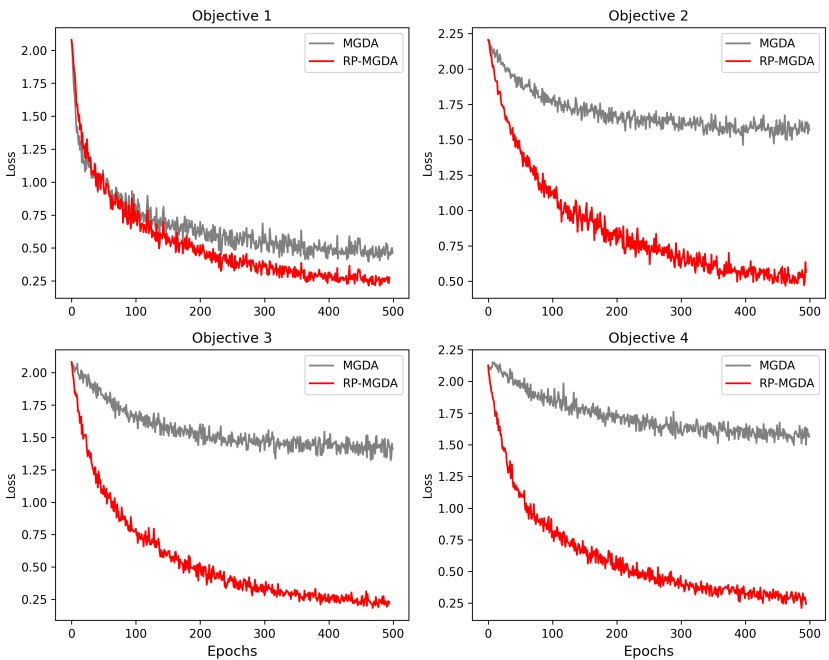

Figure 23: Personalized federated learning problem on MNIST. Comparing 4 objectives using stochastic RP-MGDA and stochastic MGDA. Report loss at the start of each epoch.

### B.3.5 PCGRAD+ AND RP-PCGRAD+

In this section, we further validate the effectiveness of Refined-Partitioning (RP) by conducting experiments on the PCGrad+ algorithm, both with and without RP, in the PFL setting. PCGrad+ is a looped version of the PCGrad algorithm (Yu et al., 2020), where the iterative projection and correction step (Lines 4-7 in Algorithm 1 of (Yu et al., 2020)) is repeated until the 'surgery' gradient $\mathbf{g}_i^{PC}$ achieves a non-negative inner product with all other gradients. This modification ensures that PCGrad+ is a common descent algorithm for all objectives even when $m \geq 3$.

For MNIST, we run 1000 epochs with lr=0.1 for both algorithms; for CIFAR-10, we run 5000 epochs with lr=0.05 and a step decay of 0.9 every 200 epochs for both algorithms. Other settings follow those of MGDA and RP-MGDA. We observe that RP-PCGrad+ outperforms PCGrad+.

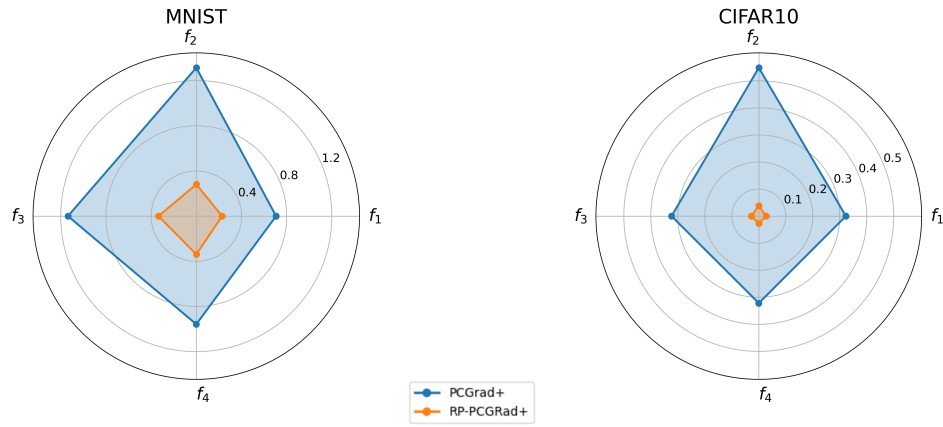

Figure 24: Radar graph of final cross-entropy losses $f_i$ in the personalized federated learning setting, using PCGrad+ and RP-PCGrad+ respectively. Lower is better.

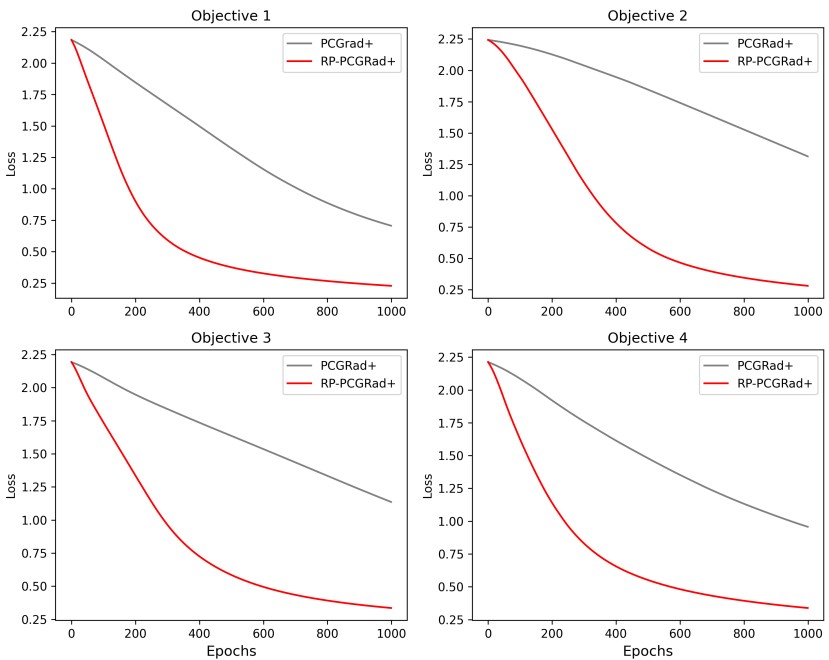

Figure 25: Personalized federated learning problem on MNIST. Comparing 4 objectives using RP-PCGrad+ and PCGrad+.

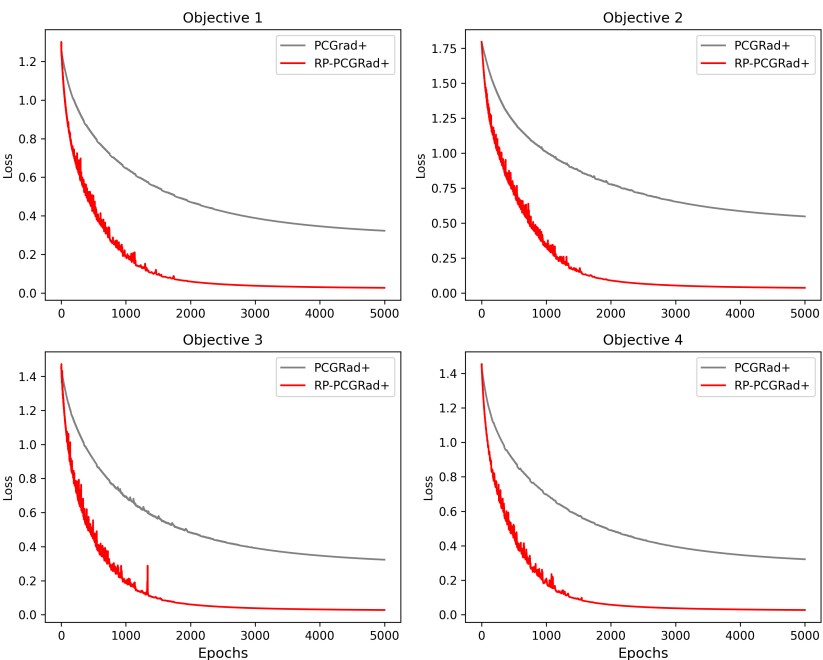

Figure 26: Personalized federated learning problem on CIFAR-10. Comparing 4 objectives using RP-PCGrad+ and PCGrad+.

### B.4 HIERARCHICAL CLASSIFICATION (THE 'LADDER' STRUCTURE)

In this section, we present detailed experiment setup.

**Data.** In addition to the original 10 classes in the CIFAR-10 dataset, we define broader categories with binary labels: ['non-living', 'living'], as well as mid-level categories: ['ground-vehicle', 'non-ground-vehicle', 'land-animal', 'non-land-animal']. These serve as coarser labels for the original classes, which include ['plane', 'car', 'bird', 'cat', 'deer', 'dog', 'frog', 'horse', 'ship', 'truck']. Consequently, each image instance in the dataset with an original label is also assigned a binary label and a mid-level label for the tasks of binary classification ($f_1$) and four-class classification ($f_2$), respectively. For instance, a 'plane' is labeled as 'non-living' 'non-ground-vehicle', while a 'deer' is labeled as 'living' 'land-animal'. We take a smaller subset of the original CIFAR-10 dataset as our training set.

**Model.** We employ a CNN architecture similar in spirit to Branch-CNN (Zhu and Bain, 2017), featuring three layers for feature extraction and three 'off-ramp' linear classifiers that branch out after the first, second, and third feature extraction layers, respectively. The first task, $f_1$, processes input through the shared feature extraction layer 1 followed by classifier 1. The second task, $f_2$, processes input through the shared feature extraction layers 1 and 2, followed by classifier 2. The third task, $f_3$, processes input through the shared feature extraction layers 1 and 2, then through the non-shared feature extraction layer 3, and finally classifier 3. See Table 3.

**Misc.** We adopt a warmstart procedure similar to the previous section by first running RP-MGDA for 5 epochs. We also employ a periodic exponential decay learning rate scheme for the experiments.

Table 3: Network architecture for hierarchical classification experiments.

| Layer Type | Configuration | Activation |
|---|---|---|
| **First Feature Extraction Layer** | | |
| Conv2D | $3 \times 32$, kernel size 5 | ReLU |
| BatchNorm2D | 32 channels | - |
| MaxPool2D | Kernel size 2, stride 2 | - |
| Conv2D | $32 \times 32$, kernel size 5 | ReLU |
| BatchNorm2D | 32 channels | - |
| MaxPool2D | Kernel size 2, stride 2 | - |
| **Second Feature Extraction Layer** | | |
| Fully Connected (FC) | $32 \times 5 \times 5 \rightarrow 384$ | ReLU |
| **Third Feature Extraction Layer** | | |
| Fully Connected (FC) | $384 \rightarrow 192$ | ReLU |
| **First Classifier** | | |
| Fully Connected (FC) | $32 \times 5 \times 5 \rightarrow 2$ | - |
| **Second Classifier** | | |
| Fully Connected (FC) | $384 \rightarrow 4$ | - |
| **Third Classifier** | | |
| Fully Connected (FC) | $192 \rightarrow 10$ | - |

### B.5 MULTI-OBJECTIVE LEARNING WITH PARTIAL INFORMATION (THE 'CHAIN' STRUCTURE)

In this section, we present detailed experiment setup.

**Data.** We conduct experiments on the California housing dataset, which consists of 20,640 samples from the 1990 U.S. Census, aiming to predict median house values. The dataset includes 8 features, ['Longitude', 'Latitude', 'AveOccup', 'Population', 'MedInc', 'HouseAge', 'AveRooms', 'AveBedrms']. We group 'Longitude' and 'Latitude' as $\mathbf{w}_1$; 'AveOccup', 'Population' and 'MedInc' as $\mathbf{w}_2$; 'HouseAge', 'AveRooms' and 'AveBedrms' as $\mathbf{w}_3$. To implement the function-variable dependency structure, we use feature masks on X during training.

**Model.** We consider a linear model with 8 weights, each corresponding to one of the 8 features in the dataset. We set the bias $b$ to be the mean of Y train.

**Misc.** We standardize each feature to have zero mean and unit variance to improve training.

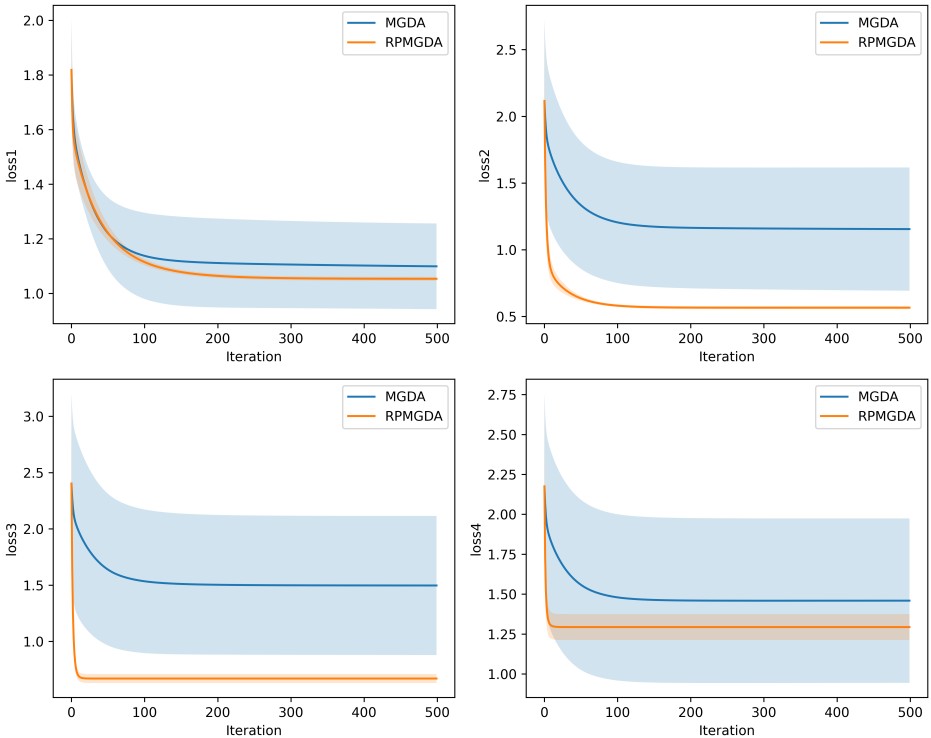

Figure 27: We calculate the mean and standard deviation of the 4 losses across the 20 trials (different seeds) for both MGDA and RP-MGDA, and plot the means, with shaded regions representing the standard deviations. Although the solution of RP-MGDA for a single trial does not dominate MGDA's. RP-MGDA achieves lower average losses for all 4 objectives and has much smaller deviations.

### B.6 JACOBI VS. GAUSS-SEIDEL UPDATE

In Section 6, we discussed the approach of cycling through the blocks of variable identified by REFINED_PARTITION for updates, also known as the 'Gauss-Seidel' update. The main RP-MGDA algorithm presented in Algorithm 1, which updates all blocks simultaneously, is referred to as the 'Jacobi' update.

In this section we empirically compare the Jacobi (default) and Gauss-Seidel versions of RP-MGDA updates, using a refined partition of 5 blocks for the PFL setting on both the MNIST and CIFAR-10 dataset. We find no significant difference overall when the number of variable updates is kept the same.

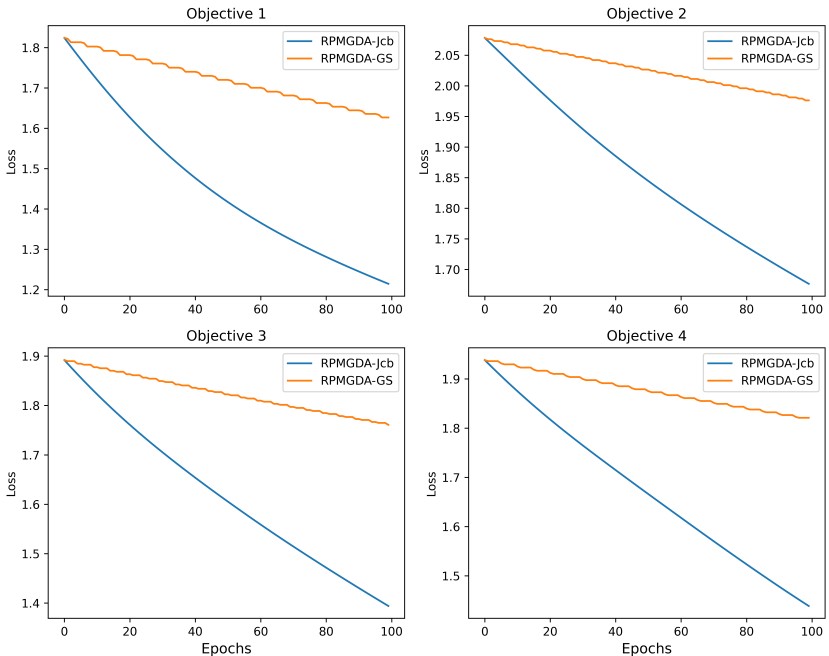

Figure 28: A look at the first 100 epochs. Gauss-Seidel update exhibits some 'zig-zags' and converges slower in terms of pure number of epochs. CIFAR-10 dataset.

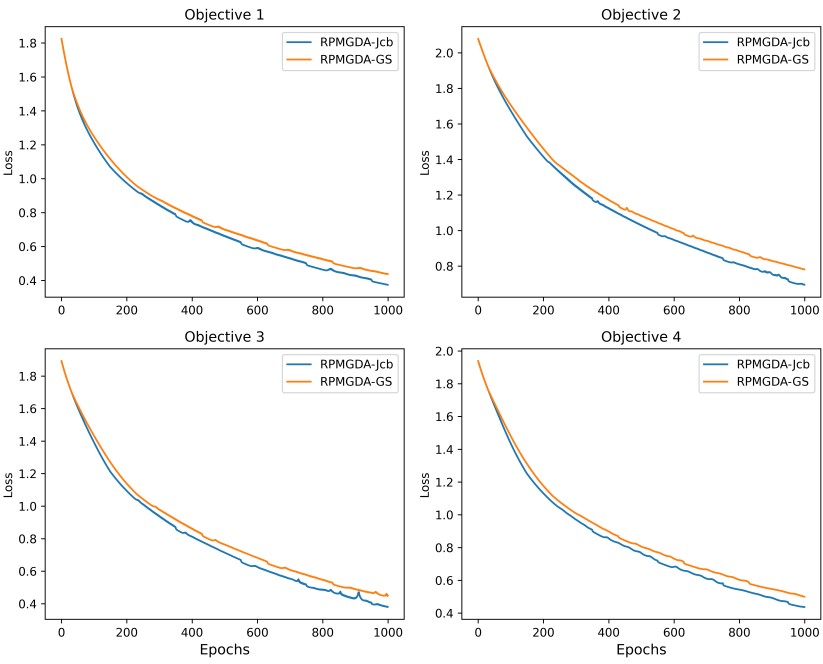

Figure 29: Overall convergence trends are similar if we plot the loss of Gauss-Seidel update every 5 epochs instead, to ensure the total number of variable updates are the same for Jacobi and Gauss-Seidel at each time stamp in the plot. CIFAR-10 dataset.

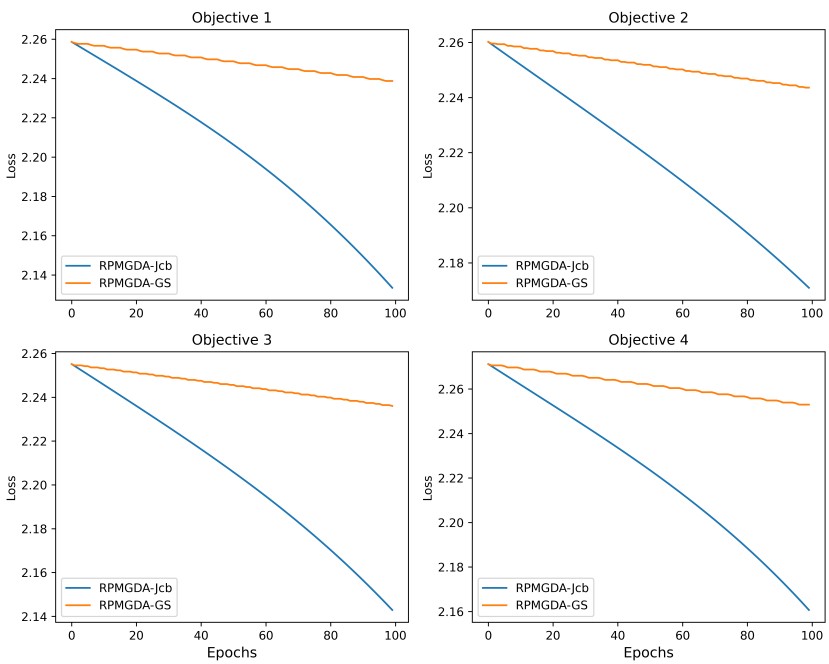

Figure 30: A look at the first 100 epochs. MNIST dataset, PFL setting.

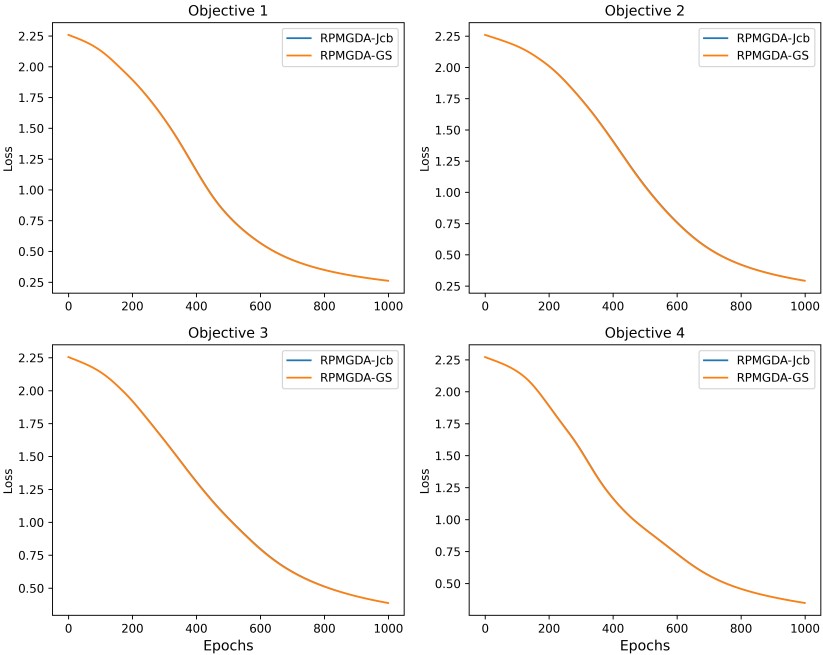

Figure 31: Overall convergence in 1000 epochs, with Gauss-Seidel update plotted every 5 epochs to ensure the total number of variable updates are the same. MNIST dataset, PFL setting.

## B.7 MISC

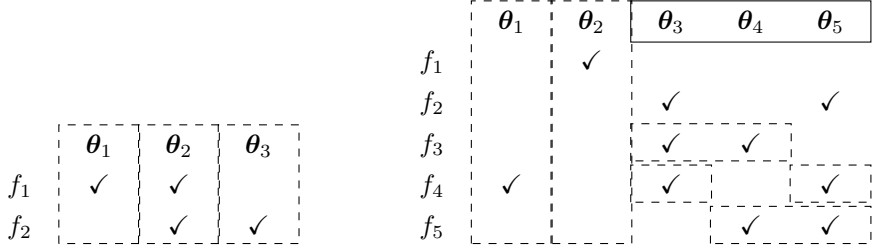

Figure 32: Variable dependency structures of synthetic toy examples in §B.1 and §B.2.

