# OpenReview forum: "Leveraging Variable Sparsity to Refine Pareto Stationarity in Multi-Objective Optimization"
_ICLR.cc/2025/Conference — ICLR 2025 Poster_

### Official Review · Reviewer_h4fS · 2024-10-23

**Soundness:** 3
**Presentation:** 3
**Contribution:** 3
**Rating:** 6
**Confidence:** 2

**Summary:**

Multi-objective optimization (MOO) have many applications in machine learning. Multiple Gradient Descent Algorithm (MGDA) is essential to solve MOO, converging to a Pareto stationary solution, which serves as a first-order necessary condition for Pareto optimality. This work demonstrates that the Pareto Stationarity (PS) has limitations when sparse function-variable dependencies exist, and to address it, they propose a concept named Refined Pareto Stationarity (RPS). With a suitable designed partitioning procedure, they propose an optimization algorithm RP-MGDA, which is effective in both the theory and experiments.

**Strengths:**

- The contributions of this paper are very clear. They provide a refined concept, and based on this, they propose a novel MOO algorithm.

- Under some convexity, they prove RPS reduces exactly to Pareto optimality, whereas the widely-used PS does not, suggesting that the new solution concept is more superior.

- A more powerful algorithm is proposed, and the advantages are supported in both the convergence and the experiments.

**Weaknesses:**

- The aim of the partition of variables should be clearly interpreted in the Introduction.

- I apologize that I am not very familiar with this topic and the relevant references. I will carefully refer to the comments of other reviewers.

**Questions:**

Is variable sparsity a common phenomenon in MOO?

---

> ### Author Response · Authors · 2024-11-19
> **Thank you for your positive feedback.**
>
> - **W1. The aim of the partition of variables should be clearly interpreted in the Introduction.**
>
>   A: Thank you for the suggestion. In the third and fourth paragraphs of the Introduction section, we mentioned that a valid partition of variables is used to refine the Pareto Stationary (PS) concept into Refined Pareto Stationarity (RPS), addressing the limitation of PS. This partition is also crucial for enabling a *proper* block-wise variant of existing gradient-based descent MOO algorithms, such as MGDA.
>
>   We agree that the aim should be more clearly articulated, and we are happy to provide the following elaborations to improve understanding:
>   - (1) The motivation for 'partitioning of variables can help' comes from the application side (e.g., the dependency structure in MTL/PFL), as demonstrated in __Example 1__. However, it is equally important to recognize that arbitrary partitioning, particularly when the dependency structure is dense, will cause more harm than good, __see Example 2__. This highlights the need to *correctly* identify a proper partition of variables that aligns with the problem's function-variable dependency structure.
>   - (2) The goal is to develop a universal approach to exploit any underlying structure and identify the finest partition that makes optimization more effective. To achieve this, we formalize the novel solution concept of Refined Pareto Stationarity (RPS), which relies on a valid partitioning of variables given by the REFINED_PARTITION() procedure. RPS offers a sharper characterization than PS, narrowing down sub-optimal PS solutions. This partitioning also enables block-wise MGDA (i.e. RP-MGDA) to fucntion without the risk of reaching degenerate solutions.
>
> - **Q1. Is variable sparsity a common phenomenon in MOO?**
>
>   A: Yes, there are structured problems in MOO with sparse dependencies (e.g., see [1]). However, the interpretation of what constitutes 'common' and 'sparse' may vary depending on individual perspectives. Please refer to the examples provided in Section 7.2 of our experiments, which illustrate dependency structures in various problems, including PFL, HC, and MOL-PI. Figure 5 provides a visual representation of these structures. Note that our framework is general and capable of handling any function-variable dependency structure, including those perceived as 'dense.' In the worst-case scenario (where the only valid partition is $\mathcal{Q}=[d]$), it reduces RPS to PS and RP-MGDA to MGDA, serving as a baseline.
>
> [1] Sener, O. and V. Koltun. (2018). "Multi-Task Learning as Multi-Objective Optimization". In: Advances in Neural Information Processing Systems.

---

> ### Author Response · Authors · 2024-12-01
> **Follow-Up on Rebuttal Discussion**
>
> Dear Reviewer,
>
> As the discussion period is nearing its end, we kindly look forward to your feedback on our responses. We wanted to share that the other reviewers who have responded have positively updated their scores after reviewing our rebuttal and the additional clarifications and experiments we provided. We would greatly appreciate your feedback to ensure a comprehensive evaluation of our work.
>
> Thank you for your time and effort.
>
> Best regards,
> The Authors

---

### Official Review · Reviewer_Vj3K · 2024-11-03

**Soundness:** 3
**Presentation:** 3
**Contribution:** 2
**Rating:** 5
**Confidence:** 3

**Summary:**

This work highlights the limitations of Pareto stationarity when dealing with sparse function-variable structures, offering compelling examples to illustrate these constraints. To overcome these challenges, they introduce a new solution concept called Refined Pareto Stationarity (RPS) and present an efficient partitioning algorithm to automatically uncover function-variable dependencies.

**Strengths:**

1. This work is presented with good writing style, where the summarized problems with detailed explanations make it easy for readers to understand the problem addressed in this article.

2. The set defined by Pareto stationarity is broader than that defined by Pareto optimal, a detail that previous algorithms based on Pareto stationarity have overlooked. This paper faces this gap and introduces algorithms to overcome this limitation, representing a significant improvement.

3.Theorems 1 and 2 provide a detailed discussion on the relationships among Refined Pareto Stationarity, Pareto optimality, and Pareto stationary points. Additionally, Theorems 3 and Corollary 1 offer theoretical guarantees for the specific convergence rate of RP-MGDA.
The proof seems solid but I have not carefully checked the whole Appendix.

4.The performance of MGDA and RP-MGDA was compared across various scenarios, demonstrating the effectiveness and versatility of RP-MGDA.

**Weaknesses:**

1.Computational Cost.  Neither MGDA nor RP-MGDA seems well-suited to large-scale machine learning problems, as stochastic variants (e.g. MOCO, MODO) are often more computationally efficient in practice. Can the RP-MGDA algorithm be adapted into a corresponding stochastic variant, and if so, would it still handle sparse parameter issues effectively after randomization?

2.Theoretical challenges. Compared to the RPS concept proposed in this paper, Pareto stationary points appear to be a clearer optimization target. What changes does the Refined Pareto Stationarity bring to the theoretical proof? What challenges arise from these changes, and how are you addressed?

3.Illustrations on Pareto front. The paper could provide a more in-depth analysis of Pareto optimal, Pareto stationarity, and Refined Pareto Stationarity. For example, the experiments could visualize the fronts corresponding to each of these concepts. Additionally, plotting the convergence trajectories of MGDA and RP-MGDA would further emphasize the effectiveness of RP-MGDA.

**Questions:**

1.Lack of practical examples. Could you provide some real-world examples of sparse function-variable structures? For instance, cases that exist in multi-objective federated learning or reinforcement learning.

---

> ### Author Response · Authors · 2024-11-19
> **Thank you for the feedback**
>
> - **W1. Stochastic variant of RP-MGDA. Can the RP-MGDA algorithm be adapted into a corresponding stochastic variant, and if so, would it still handle sparse parameter issues effectively after randomization?**
>
>   A: Thank you for the questions.
>   - It is natural to propose the stochastic counterpart of RP-MGDA (i.e. simply replacing deterministic gradients with stochastic ones). However, analyzing this extension rigorously (e.g., convergence properties) requires more effort. Although implementing a stochastic version of MGDA is straightforward, there is a developing body of literature addressing various potential issues, such as biased descent directions and the need for additional assumptions to establish convergence ([1] [2]). Thus, we see this as a promising direction to explore.
>   - Randomness in gradients (e.g., from mini-batch training) is orthogonal to the function-variable dependency structure and does not interfere with it. Importantly, the generality of our proposed RPS *solution concept* is not tied to any specific algorithm, so the effectiveness of the refined partitioning approach should remain unaffected by gradient noise. However, there are potential challenges to consider:
>     - (i) In the stochastic setting, the descent property may not hold in every iteration, potentially undermining RP-MGDA's effectiveness as a post-processing refinement algorithm (Figure 4 Left). It also challenges the utility of the commonly used 'compact sublevel set' assumption.
>     - (ii) Addressing the bias in stochastic descent directions without introducing additional dependencies between functions and variables could require extra caution. Note that for Refined Partitioning, it is crucial to preserve sparsity as much as possible to ensure the most effective variable partitioning.
>   - We have added experiments using stochastic gradients for both MGDA and RP-MGDA; see Appendix B (Pages 24 and 25, Figures 21–23). The empirical results continue to validate the superiority of the refined partitioning approach, with the benefits being arguably even greater in the stochastic setting. However, we note that RP-MGDA and MGDA no longer guarantee descent in every iteration (see Figures 22 and 23). The performance of both methods drops slightly, potentially due to bias in the descent direction, which supports our conjecture regarding the associated challenges.
>
> [1] Fernando, H. et al.  (2023). "Mitigating Gradient Bias in Multi-objective Learning: A Provably Convergent Approach". In International Conference on Learning Representations.
>
> [2] Liu, S., & Vicente, L. N. (2021). "The stochastic multi-gradient algorithm for multi-objective optimization and its application to supervised machine learning". Annals of Operations Research, vol. 339, pp. 1119–1148.
>
> - **W2. Theoretical benefits and challenges of RPS**
>
>   A: Thank you for the questions.
>   - (1) As pointed out in Section 5 (see Lemma 2, Theorem 1 and Figure 3), RPS provides a sharper characterization than PS (PO $\subseteq$ RPS $\subseteq$ PS), meaning it is 'closer' to the desired Pareto Optimal set and narrows down sub-optimal PS solutions. Theorem 2 (also empirically verified in Section 7.2.3 and Appendix A, Example 3) further demonstrates that RPS is guaranteed to achieve PO under more relaxed assumptions, whereas PS is not. Together, these results indicate that RPS is a more effective proxy to pursue than PS.
>   - (2) The challenge lies in finding the *correct* partitioning of variables, which is crucial because an overly fine-grained partition may result in Generalized Pareto Stationary (GPS) with respect to that partition failing to imply Pareto Stationary (PS), and block-wise MGDA may fail (see Example 2). Conversely, an overly coarse partition yields no benifit and suffers from the drawback of PS (see Example 1).
>
>     We successfully address this challenge by proposing REFINED_PARTITION() procedure which leverages the function-variable dependency structure (represented as bipartite graph) through cycle detection and variable merging, to identify a *valid* refined partition. This systematic procedure is theoretically guaranteed by Theorem 1, whose proof relies on the final bipartite graph being acyclic.

---

> ### Author Response · Authors · 2024-11-19
>
> - **W3. Illustrations on Pareto front**
>
>   A: Thank you for the suggestion.
>
>   The Pareto front (i.e., the set of Pareto Optimal points) is difficult, if not impossible, to determine for non-convex high-dimensional problems, which are common in modern MOO problems. Even for simple convex problems where Pareto fronts are attainable, visualizing them becomes hardly practical for more than two objectives. Therefore, for our experiments on real benchmark datasets, plotting the Pareto front is not feasible.
>
>   However, for convex and 2-objective problems, we did compare the solution fronts of MGDA and RP-MGDA, as shown in Appendix B, Figure 10. Regarding the trajectory of convergence, if referring to the trajectory in the objective space ($\mathbb{R}^m$), visualization is not possible for $m \geq 3$. Instead, we track the trajectories of *each* objective value (i.e., loss curves) for RP-MGDA and MGDA in e.g., Figure (7,8,14-20). Additionally, we provide the convergence of the norm $\|\mathbf{d}\|$ of the updating directions for MGDA and RP-MGDA in e.g. Figure 12 (Right).
> - **Q1. Lack of practical examples**
>
>   A: In Experiment Section 7.2, we explored three examples with different function-variable dependency structures, derived from real-world applications:
>
>     - Personalized federated learning [1,2] (MTL structure)
>     - Hierarchical classification [3,4] (Ladder structure)
>     - Multi-objective learning with partial information [5] (Chain structure)
>
>   The illustrations of the problem structures and the corresponding variable partitioning are shown in Figure 5, with detailed setups for these experiments discussed in Section 7.2 and Appendix B. Experimental results have demonstrated the effectiveness of Refined Partitioning. While it is clear that we cannot exhaustively cover all real-world MOO problems with non-trivial function-variable dependency structures in this paper, the examples presented serve as a ‘proof of concept’. They provide a foundation for practitioners to apply refined partitioning to other realistic MOO problems of interest with non-trivial structures.
>
>   Note that our framework is general and capable of handling any function-variable dependency structure, including those perceived as 'dense.' In the worst-case scenario (where the only valid partition is $\mathcal{Q}=[d]$), it reduces RPS to PS and RP-MGDA to MGDA, serving as a baseline.
>
> [1] Liang, Paul Pu, et al. "Think locally, act globally: Federated learning with local and global representations." arXiv preprint arXiv:2001.01523 (2020).
>
> [2] Tan, Alysa Ziying, et al. "Towards personalized federated learning." IEEE transactions on neural networks and learning systems 34.12 (2022): 9587-9603.
>
> [3] Zhu, Xinqi, and Michael Bain. "B-CNN: branch convolutional neural network for hierarchical classification." arXiv preprint arXiv:1709.09890 (2017).
>
> [4] Seo, Yian, and Kyung-shik Shin. "Hierarchical convolutional neural networks for fashion image classification." Expert systems with applications 116 (2019): 328-339.
>
> [5] Liu, Yang, et al. "Vertical federated learning: Concepts, advances, and challenges." IEEE Transactions on Knowledge and Data Engineering (2024).

---

> ### Author Response · Authors · 2024-11-30
>
> Dear reviewer,
>
> The discussion period is due on Dec 2nd, could you please provide some feedback or continue the discussion if you have further questions?
> We really appreciate your time and effort in reviewing this paper. Your response is important to us.
>
> Best regards,
> The authors.

---

### Official Review · Reviewer_EgBd · 2024-11-03

**Soundness:** 2
**Presentation:** 2
**Contribution:** 2
**Rating:** 6
**Confidence:** 3

**Summary:**

This paper studies the multi-objective optimization problem and reveals the limitation of the widely used metric, Pareto stationarity. Accordingly, the authors propose the refined Pareto stationarity sandwiched between Pareto stationarity and Pareto optimality. Then, the authors verify the benefit of their RP-MGDA empirically and theoretically.

**Strengths:**

1. This paper has a solid standing point that Pareto stationarity could be short in complex and sparse settings.
2. Experiments explore multiple variable dependency structures.

**Weaknesses:**

1. Some important parts are confusing in the paper. 1) How can we get the function-variable dependency structure? This is important because the dependency structures vary in different settings. 2) Algorithm REFINED_PARTITION is not clear. A more detailed explanation of it should be added. In the current version, I cannot say I fully understand it.
2. Since I did not fully understand the partition, what does "finer" or "coarser" partition mean?
3. According to the illustration of the function-variable dependency structure and the experimental setup in lines 467-468, I understand this function-variable dependency structure as a dependency between objectives and "specific layers" in the model. However, I do not feel this setup fits my sense. From my point of view, a dependency should be between objectives and some part of weights in each layer if the model is sparse (Correct me if I am wrong). A simple example would be this: suppose we have m objectives and the model is a n-layer neural network. Then the objective $i$ is dependent on weights $[w_{1, i},..., w_{n,i}]$ where the first index of weights represents the layer index and the second index of weight represents the objective index. Why do the authors consider the dependency in your way? Also, if consider my settings, how to do the partition?
4. Though this paper has considered multiple dependency structures in the experiments, tasks, and datasets are a bit easy. Previous related papers compare the performance of Cityscapes, NYU-v2, CelebA, etc. Do authors consider checking these experiments? In addition, this paper only compares MGDA and RP-MGDA, which is short in the number of methods. Lastly, can the partition be added to other MGDA-based methods?

**Questions:**

Please check the weaknesses.

---

> ### Author Response · Authors · 2024-11-19
> **Thank you for your detailed feedback. We have addressed your questions below and provided an important clarification on 'sparsity' in our response to W3.**
>
> - **W1. Some important parts are confusing in the paper. 1) How can we get the function-variable dependency structure? 2) Procedure REFINED_PARTITION() is not clear.**
>
>   A: Thank you for the questions. We aim to provide as much clarification as possible below, and please let us know if our answer addresses your concern.
>     - (1) The function-variable dependency structure is part of the problem definition: when implementing any objective function, it is already clear which variable(s) it depends on—or does not. In other words, the adjacency matrix $A$  representing the function-variable dependencies of the problem is given. The challenge is whether and how to effectively exploit this given structure for better optimization. In synthetic examples, the dependencies are explicitly defined by the function expressions. In empirical problems, the dependency structure is inherently dictated by the problem setup----informally, by what is shared and what is not. For instance, in our Personalized Federated Learning setup, a global neural network model is shared across users, whereas each user's local personalization model is unique to their respective objective, see Section 7.2.1 and Appendix B.3.1 for details.
>
>       For an intuitive understanding, __we recommend referring to Example 1 and 2 in Section 4, particularly Figure 2__. Furthermore, please refer to Section 7.2 where we provide three detailed examples demonstrating various dependency structures  in real-world applications, __see Figure 5 for a graphical illustration__.
>
>     - It is worth noting that what constitutes 'variables' is flexible and context-dependent; it can be fine-grained or coarse-grained. For instance:
>       - In Example 1 and 2, variables are scalars.
>       - In Personalized Federated Learning (Section 7.2.1), variables are personalized neural network models and a global neural network model.
>       - In Hierarchical Classification (Section 7.2.2), variables are feature extraction layers of the BranchNet.
>       - In MOL-PI (Section 7.2.3), variables are (grouped) linear regression weights.
>     - (2) Briefly, REFINED_PARTITION() is a repeated process that finds any cycle (if exists) in the underlying bipartite graph (see Eq (12) for the adjacency matrix A representation of the bipartite graph), then groups (aka. 'merges') all variables appeared in that cycle together and thus contracts the graph. The process is repeated until the final graph is acyclic. __Please refer to Figure 1 for an illustration__. We have also added the pseudo-code for REFINED_PARTITION() in Appendix A, Page 16, Line 848-863. Below, we provide a simplified plain-language description of the procedure.
>
>
>       **Algorithm: REFINED_PARTITION()**
>       **Input:** A - Adjacency matrix representing the function-variable dependencies (bipartite graph).
>       **Output:** $\mathcal{P}$ - Refined partition of variables.
>       1. Initialize $\mathcal{P}$ ← { {$w_1$}, {$w_2$}, ..., {$w_d$} } (each variable starts in its own group).
>       2. While a cycle exists in the bipartite graph represented by A:
>
>           2.1. Detect a cycle $C$ in the bipartite graph.
>
>           2.2. Merge all variable nodes in $C$ into a single group (e.g., merge {$w_1$} and {$w_2$} into {$w_1, w_2$}).
>
>           2.3. Update A to reflect the contraction of the graph.
>
>           2.4. Replace the original groups in $\mathcal{P}$, corresponding to the variables in $C$, with the merged group.
>
>       3. Return $\mathcal{P}$.
>
>       On a high level, REFINED_PARTITION() is a key systematic approach we propose to identify a *valid* partition of variables into 'blocks' (referred to interchangeably as 'groups' or 'subsets') for a given problem. This partition enables a sharper characterization (i.e. RPS) than Pareto stationarity, and allows for block-wise MGDA (i.e. RP-MGDA) to function without the risk of converging to degenerate solutions (see Example 2 for a counterexample).
>
>       Otherwise, if a partition is too fine, Generalized Pareto Stationary with respect to that partition is not guaranteed to imply Pareto Stationary and block-wise MGDA may fail (see Example 2); conversly, if a partition is too coarse, no benifits are gained, and we will suffer from the drawback of Pareto Stationarity (see Example 1).

---

> ### Author Response · Authors · 2024-11-19
>
> - **W2: what does "finer" or "coarser" partition mean?**
>
>   A: Thank you for this question. A partition $\mathcal{P}$ of a set $\\{1, 2, \ldots, d \\} $ is a collection of subsets $P \subseteq \\{1, 2, \ldots, d \\}$ such that every element is included in *exactly* one subset $P \in \mathcal{P}$. We call a partition $\mathcal{P}$ finer than another partition $\mathcal{Q}$ (equivalently, $\mathcal{Q}$ is coarser than $\mathcal{P}$), if for any $Q \in \mathcal{Q}$ there exists a $P \in \mathcal{P}$ such that $P \subseteq Q$; see Remark 2 in Appendix A (line 742-744). In other words, a finer partition consists of more subsets, with each subset being smaller. For example, let $$ \mathcal{P} = \\{ \\{1\\}, \ldots, \\{d\\} \\}, ~~\mathcal{Q} = \\{\\{1, \ldots, d\\} \\}.$$
>   Then, $\mathcal{P}$ is finer than $\mathcal{Q}$. In fact, this $\mathcal{P}$ is finest and this $\mathcal{Q}$ is coarsest (referred to as the trivial partition).
>
>   In this terminology, MGDA and other works in MOO [1] are typically formulated using the coarsest variable partition $\mathcal{Q}$, while we demonstrate in this paper that a *proper* finer partition of variables (when it exists) is strictly superior to using the coarsest partition.
>
>   - There are two perspectives to understand the refined-partition idea: top-down and bottom-up.
>     - The __top-down perspective__ starts with the coarsest partition where Theorem 1 trivially holds (e.g., MGDA in Example 1), and then identifies the finest-grained partition possible while ensuring Theorem 1 still holds. This approach refines the traditional PS concept, where PS corresponds to the special case of the trivial partition $\mathcal{Q}=[d]$. This perspective is more suitable for theoretical purposes.
>     - The __bottom-up perspective__ starts with the default of treating every variable separately (e.g, coordinate-wise MGDA in Example 2), where Theorem 1 is not guaranteed. Variables must then be iteratively merged based on function dependencies, until a coarser partition is reached where Theorem 1 holds. This perspective is more suited for constructive purposes.
>
> [1] Fernando, H. D., H. Shen, M. Liu, S. Chaudhury, K. Murugesan, and T. Chen (2023). “Mitigating Gradient Bias in Multi-objective Learning: A Provably Convergent Approach”. In: International Conference on Learning Representations.

---

> ### Author Response · Authors · 2024-11-19
>
> - **W3. Why in lines 467-468, function-variable dependency structure is between objectives and specific layers? Shouldn't the dependency be between objectives and part of weights in each layer for a sparse model? Discuss the simple example I proposed and how to do the partition.**
>
>   A: Thank you for the question.
>   - As noted in our previous response to W1, the choice of 'variables' is flexible and context-dependent, ranging from fine-grained to coarse-grained. In Hierarchical Classification (Section 7.2.2), we consider variables to be feature extraction layers in BranchNet, as this best captures the relevant dependency structure for this problem setup.
>   - It is certainly possible to define variables differently—for example, by treating each individual weight as a variable. However, following the REFINED_PARTITION() procedure, one will find that these weights will need to be merged due to their inclusion in certain cycles (in the exact same way as illustrated in Example 2, Figure 2 Right). This outcome, unsurprisingly, aligns with common sense and the underlying principles of our framework, which, in turn, supports the soundness of the REFINED_PARTITION() procedure.
>   - __Further clarification on 'Sparsity'.__  Below, we want to clearly distinguish between the following two concepts, as this distinction is crucial for accurately understanding our paper:
>     - (1) __Sparse model weights:__ This refers to models where many of the final optimal weights are zero after training is complete (e.g., in Lasso regression). Importantly, during training, the objective function still depends on all weights.
>     - (2) __Sparse function-variable dependency:__ This concept pertains to the structure of the problem itself, observed externally, where 'variables' typically correspond to neural network modules in empirical deep learning problems. The dependency structure is naturally determined even before training begins.
>
>     Our paper specifically focuses on the __second__ concept, and we do not claim or imply sparsity in model weights in our work. Next, we walk through the Hierarchical Classification (HC) example to further explain the concept:
>
>     In our HC setup, the dependency structure is inherently determined by the architecture of BranchNet and where we put the 'off-ramp' classifiers. Each 'off-ramp' classifier corresponds to a different classification objective (from easy to difficult). For example, the first objective $f_1$ uses only the representation produced after FEX-Layer 1 and does not interact with the 2nd or 3rd layers. Consequently, it depends solely on $\mathbf{w}_1$. Similarly, $f_2$ does not interact with the 3rd layer, thus depending on $\mathbf{w}_1$ and $\mathbf{w}_2$, but not $\mathbf{w}_3$. Finally, $f_3$ uses the output after FEX-Layer 3, thus depending on all the modules mentioned before, i.e. $\mathbf{w}_1$, $\mathbf{w}_2$ and $\mathbf{w}_3$.
>
>     It is now clear why the dependency structure is represented as shown in Figure 5 (Middle). We can then use REFINED_PARTITION() to explore whether a finer partition can be identified for this problem structure. While this Ladder structure is not particularly 'sparse', we can still find a finer partition.
>   - We appreciate the interesting setting you have proposed and would be happy to discuss it further.
>     - First, we follow your proposed dependency literally, where each objective $f_i$ depends on $[w_{1,i},\ldots,w_{n,i}]$ only. Then in our refined partition framework, every variable (treating each weight $w_{k,i}$ as a variable) can be optimized separately using gradient descent, with the corresponding refined partition being the finest partition. Note that for this dependency structure, even MGDA is not needed since no two objectives share the same variable. Indeed, this is essentially equivalent to $m$ separate single-objective optimization problems since no variable is shared at all.
>     - This dependency structure seems somewhat unconventional, especially regarding how the proposed dependency might be enforced. For example, if a function, say $f_i$, depends on the output of the $n$-th layer, it would naturally involve all weights in the preceding layers. It is unclear how it could depend solely on $w_{k,i}$ for the $k$-th layer.
>     - We would like to reiterate our earlier clarification that 'sparsity' refers to sparse function-variable dependencies, not sparse model weights, to ensure we are aligned and avoid any potential misunderstandings.
>
>     We kindly ask for clarification on the context of the objective functions being discussed or an example to help us better understand your perspective. If our current interpretation is incorrect, we would appreciate more details—for instance, whether the objective functions are loss functions, what representations they depend on, or an example of how the proposed dependency could hold true. We would be happy to discuss further based on this additional information.

---

> ### Author Response · Authors · 2024-11-19
>
> - **W4. Though this paper has considered multiple dependency structures in the experiments, tasks, and datasets are a bit easy. Previous related papers compare the performance of Cityscapes, NYU-v2, CelebA, etc. Do authors consider checking these experiments? In addition, this paper only compares MGDA and RP-MGDA, which is short in the number of methods. Lastly, can the partition be added to other MGDA-based methods?**
>
>   A: Thank you for your comments. We would like to address your points as follows:
>   - Regarding the choice of tasks and datasets: While we acknowledge that most multi-task learning papers have explored datasets such as Cityscapes, NYU-v2, and CelebA, our primary focus in this work is on demonstrating the effectiveness of the refined partitioning framework for multi-objective optimization across *various* dependency structures, rather than the complexity of specific tasks or datasets for multi-task learning. The dependency structures we selected were designed to align closely with the theoretical contributions of our paper and to serve as clear, illustrative examples. Also note that the MTL dependency structure in [2] is essentially explored through our PFL experiments which share the same structure. Expanding to more challenging datasets like Cityscapes or NYU-v2, while valuable, is not essential to validate our primary claims. However, we appreciate the suggestion and do consider incorporating such experiments in the revision or as part of an extended study if they are found to provide additional meaningful insights.
>   - On the number of methods compared: The goal of our experiments is to explore and validate the effectiveness of the refined partitioning approach under various dependency structures. Since we use MGDA as a solid starting point and rigorously analyze its refined-partitioning variant (e.g., Theorem 3), this comparison is the most fair and relevant for evaluating our contributions. In contrast, directly comparing RP-MGDA to methods like PCGrad [3] would not be as fair or meaningful in this context. On the other hand, we do agree that additional comparisons, such as evaluating another method (e.g., PCGrad) with and without refined partitioning, could further enhance confidence in our RP approach. __In the revision, we included experiments comparing PCGrad+ (a modified version of PCGrad ensuring it is a common descent algorithm) with and without Refined Partitioning.__ The results demonstrate that RP improves empirical convergence and solution quality (see Appendix B.3.5, Pages 25–26).
>   - On adding partitions to other MGDA-based methods: Thank you for this question. The refined partition approach and the RPS solution concept are designed to be general and could be extended to other gradient-based descent methods that converge to Pareto Stationarity (see the discussion in Section 6 Line 313-318). We agree with you that studying these extensions is a valuable and interesting direction, but it goes a bit beyond the scope of this work. We extensively study the widely-used foundational MGDA algorithm as a starting point, and we hope this work will inspire further research in this direction.
>
> [2] Sener, O. and V. Koltun. (2018). "Multi-Task Learning as Multi-Objective Optimization". In: Advances in Neural Information Processing Systems.
>
> [3] Yu, Tianhe, et al. "Gradient surgery for multi-task learning." Advances in Neural Information Processing Systems 33 (2020): 5824-5836.

---

> ### Author Response · Authors · 2024-11-30
>
> Dear reviewer,
>
> The discussion period is due on Dec 2nd, could you please provide some feedback or continue the discussion if you have further questions?
> We really appreciate your time and effort in reviewing this paper and want to make sure that our work is correctly understood. We spent the most time trying to clarify and address your questions, and your response will be of great importance to us.
>
> Best regards,
> The authors.

---

> > ### Comment · Reviewer_EgBd · 2024-12-01
> >
> > Thanks for the detailed answers. I have raised my score.

---

### Official Review · Reviewer_MUMM · 2024-11-03

**Soundness:** 3
**Presentation:** 4
**Contribution:** 3
**Rating:** 8
**Confidence:** 3

**Summary:**

By leveraging refined variable partitioning, this work introduces a novel solution concept, Refined Pareto Stationarity (RPS), and a variant of the Multiple Gradient Descent Algorithm (MGDA), termed RP-MGDA, to address limitations of Pareto stationarity and MGDA in multi-objective optimization. RPS provides a sharper characterization than Pareto stationarity, and RP-MGDA is proven to converge to RPS. Comprehensive experiments demonstrate the superior performance of RP-MGDA over vanilla MGDA.

**Strengths:**

$\textbf{S1:}$ This work reveals limitations of Pareto stationarity and  Multiple Gradient Descent Algorithm (MGDA), commonly used in multi-objective optimization, particularly when the function-variable dependency structure is sparse. The authors provide illustrative examples showing that a variable partitioning scheme is crucial for addressing these limitations.

$\textbf{S2:}$ Building on this insight, the authors introduce a novel solution concept, Refined Pareto Stationarity (RPS), which corresponds to the finest (or refined) variable partition aligned with the function-variable dependency structure. RPS is shown to be a sharper characterization than Pareto stationarity, from both the necessary and sufficient conditions perspective for Pareto optimality.

$\textbf{S3:}$ Utilizing this refined variable partitioning, the authors propose a variant of MGDA, termed RP-MGDA, which converges to RPS and is theoretically more efficient. Empirical results further demonstrate the superior performance of RP-MGDA compared to vanilla MGDA.

**Weaknesses:**

$\textbf{W1:}$ If the full gradient is used in (11) of Definition 4 (Generalized Pareto Stationarity), then any Pareto stationary point with respect to any variable partition would also be Pareto stationary with respect to the trivial partition. This suggests that the current version of Definition 4 is not consistent with the motivation behind RPS or Algorithm 1 (RP-MGDA). It seems that replacing the full gradient $\nabla \bf{f}^{P_j}$ with the partial gradient $\nabla_{\bf{w}_{P_j}} \bf{f}^{P_j}$ could address this issue. Please let me know if my understanding is incorrect, as I may adjust my ratings based on your response.

$\textbf{W2:}$ Since RP-MGDA and the vanilla MGDA have a similar convergence rate, the authors discuss the computational complexity of RP-MGDA at the end of Section 6 and state that RP-MGDA is theoretically cheaper than the vanilla MGDA, aside from the one-time overhead in line 1. However, the supporting argument lacks detail. A more thorough complexity analysis, including explicit computational cost comparisons for solving the dual subproblem in both algorithms, would be beneficial in clarifying the computational savings.

**Questions:**

Apart from the questions raised in the Weaknesses section, I have a few additional questions:

$\textbf{Q1:}$ In Theorem 3, it appears that $\eta \leq \min_i \frac{1}{L_i}$ may be insufficient, as it seems that (32) cannot be derived directly from (31) using (33). Could you please verify (32) in the proof of Theorem 3 on Page 16? If my observation is correct, one alternative might be to set $\eta \leq \min_i \frac{1}{2L_i}$ in Theorem 3 and adjust the proof accordingly.

$\textbf{Q2:}$ In the experiments of Section 7.2, is a stochastic variant of RP-MGDA used? Additionally, when ReLU is applied, what type of derivative is used in the implementation of RP-MGDA? Could this method be extended to a stochastic setting, and what significant challenges might arise in making such an extension?

---

> ### Author Response · Authors · 2024-11-19
> **Thank you for your insightful comments and we answer your questions below.**
>
> - **W1. For Definition 4 (11) in the main paper, should we use partial gradient?**
>
>   A: Yes, your understanding is correct, and thank you for pointing out this notation issue. To clarify, here we did intend to use *partial* gradients, not full gradients, in Definition 4, since we handle each block of variables independently (as shown in Algorithm 1 Line 4; and our proof of Lemma 2 in the Appendix). When we wrote $\nabla \mathbf{f}^{P_j}(\mathbf{w} _ {\star})$, we actually meant $\nabla_{\mathbf{w}_{P_j}} \mathbf{f}^{P_j}(\mathbf{w} _ {\star})$. We apologize for the confusion and we have updated this notation in the revision (highlighted in purple).
>
> - **W2. Discussion of RP-MGDA computational complexity lacks detail.**
>
>   A: Thank you for the suggestion. There are multiple ways to solve the dual problem in existing works, so the actual complexity depends on the algorithm used. As in the submission, we use $m$ for the number of objectives and $d$ for the dimension of the gradient.
>   - interior point method (IPM) for convex QP [1]:  $O(m^3+m^2d)$ (per-iteration cost + PSD matrix computation)
>   - Nesterov's accelerated gradient:  $O(md+m\log m)$ (grad computation + projection)
>   - Frank-Wolfe: $O(md)$ (grad computation)
>
>   (We remind that above is the per-iteration complexity and normally we need to run multiple iterations.)
>
>   We note that for RP-MGDA, $m$ is typically smaller as fewer objectives are likely to be considered for each block of variables, due to sparsity. From a parallel execution perspective, $d$ is also reduced, whereas in a serial environment, $d$ essentially remains the same. For example, in a general 'Chain' structure with $m$  functions and $m-1$ variables, as considered in Section 7.2.3 (see Figure 5 Right for an illustration when $m=4$): $m$ is reduced to a constant value $2$, and $d$ is reduced to $\frac{1}{m-1}$ of its original size.
>
> [1] Potra, F.A., & Wright, S.J. (2000). "Interior-point methods". Journal of Computational and Applied Mathematics, 124(1-2), 281-302.
>
> - **Q1. Verify (32) in the proof of Theorem 3 on Page 16**
>
>   A: Thank you for the question. In the revision, we've added further justification for this step. We recall that for a 1-strongly convex (not necessarily differentiable) function $h$ (i.e., $h-\tfrac12 \|\cdot\|^2$ is convex), the following inequality holds, see e.g. Nesterov (2018, Corollary 3.2.3, p. 210): $$\forall x, ~~ h(x) \geq h(x^*) +\tfrac{1}{2} \|x-x^*\|^2,$$
>   where $x^*$ is the unique minimizer of $h$.
>
>   We apply the above inequality to the definition of $\mathbf{d} _ {P_j}^t$:
>   $$\mathbf{d} _ {P_j}^t = \mathop{\mathrm{argmin}} _ {\mathbf{d} _ {P_j}} ~ \max _ {i \in F_j} ~ \left[\nabla _ {\mathbf{w} _ {P_j}}f_i(\mathbf{w}^t) \cdot  \mathbf{d} _ {P_j} + \tfrac{1}{2} \|\mathbf{d} _ {P_j}\|^2\right].
>   $$
>   Indeed, the objective above is 1-strongly convex in $\mathbf{d} _ {P_j}$ (note that the norm term does not depend on $i$, so it can be pulled out of the max operator). Setting $x = \mathbf{0}$ and $x^*= \mathbf{d} _ {P_j}^t$ we obtain:
>
>   \begin{align}
>   0 \geq \max _ {i \in F_j} ~ \left[\nabla _ {P_j}f_i(\mathbf{w}^t) \cdot \mathbf{d} _ {P_j}^t + \tfrac{1}{2} \|\mathbf{d} _ {P_j}^t\|^2\right] + \tfrac12 \|\mathbf{0} - \mathbf{d} _ {P_j}^t \|^2.
>   \end{align}
>   Thus, rearranging we obtain (32) from (31), i.e. Line 815 to 819 in Appendix A.

---

> ### Author Response · Authors · 2024-11-19
>
> - **Q2. implementation details on RP-MGDA: did we use stochastic variant in Section 7.2 benchmark experiments? What is the gradient for ReLU? Discuss the potential extension and challenges to stochastic setting.**
>
>   A: Thank you for this comment. We use the deterministic versions of both MGDA and RP-MGDA in our benchmark experiments to align with our methodology and theoretical framework. This ensures the results are not confounded by other less relevant factors.
>   - Extension to stochastic setting and challenges:
>   It is natural to propose the stochastic counterpart of RP-MGDA (i.e. simply replacing deterministic gradients with stochastic ones).
>   However, analyzing this extension (e.g., convergence properties) requires substantial effort.
>     - Although implementing a stochastic version of MGDA is straightforward, there is a developing body of literature addressing various potential issues, such as biased descent directions and the need for additional assumptions to establish convergence ([2] [3]).
>     - We see this as a promising direction to explore, especially given the generality of the proposed RPS solution concept, which is not limited to any specific algorithm. However, there are challenges to consider:
>
>       (i) In the stochastic setting, the descent property may not hold in every iteration, potentially undermining RP-MGDA's effectiveness as a post-processing refinement algorithm (Figure 4 Left). In particular, the commonly used 'compact sublevel set' argument (due to descending of the algorithm) no longer applies.
>
>       (ii) Addressing the bias in stochastic descent directions without introducing additional dependencies between functions and variables could require extra caution. Note that for Refined Partitioning, it is crucial to preserve sparsity as much as possible to ensure the most effective variable partitioning.
>   - We have added experiments using stochastic gradients for both MGDA and RP-MGDA; see Appendix B (Pages 24 and 25, Figures 21–23). The empirical results continue to validate the superiority of the refined partitioning approach, with the benefits being arguably even greater in the stochastic setting. However, we note that RP-MGDA and MGDA no longer guarantee descent in every iteration (see Figures 22 and 23). The performance of both methods drops slightly, potentially due to bias in the descent direction, which supports our conjecture regarding the associated challenges.
>   - We use the standard PyTorch package for automatic differentiation, where the derivative of ReLU is defined to be $0$ at $0$ (see [here](https://discuss.pytorch.org/t/gradient-of-relu-at-0/64345)). This is standard in deep learning, including MGDA-related ones (e.g., MTL-MOO [4], PCGrad [5]), although technically ReLU is not differentiable at 0 (a subgradient can be defined). A rigorous analysis may require multi-objective subgradient methods, e.g., [6], which would be an interesting future extension. We have also added additional experiments with ELU replacing ReLU as activation (ELU is differentiable everywhere), and verified that the experiment results and conclusions are similar (see Appendix B, Figure 18-20).
>
>
> [2] Fernando, H. et al.  (2023). "Mitigating Gradient Bias in Multi-objective Learning: A Provably Convergent Approach". In International Conference on Learning Representations.
>
> [3] Liu, S., & Vicente, L. N. (2021). "The stochastic multi-gradient algorithm for multi-objective optimization and its application to supervised machine learning". Annals of Operations Research, vol. 339, pp. 1119–1148.
>
> [4] Sener, O. and V. Koltun. (2018). "Multi-Task Learning as Multi-Objective Optimization". In: Advances in Neural Information Processing Systems.
>
> [5] Yu, T., Kumar, S., Gupta, A., Levine, S., Hausman, K., & Finn, C. (2020). "Gradient Surgery for Multi-Task Learning". Advances in Neural Information Processing Systems, 33, 5824-5836.
>
> [6] Da Cruz Neto, J.X., Da Silva, G.J.P., Ferreira, O.P. et al. (2013). "A subgradient method for multiobjective optimization". Comput Optim Appl 54, 461–472.

---

> > ### Comment · Reviewer_MUMM · 2024-11-23
> > **Thank you for the response. I raise my rating to 8.**
> >
> > Thank you for your response and the additional experimental results, which effectively addressed my concerns. I appreciate the refined partition approach and the introduction of the Refined Pareto Stationarity solution concept. Accordingly, I raise my rating to 8 (accept, good paper).

---

> > > ### Author Response · Authors · 2024-11-23
> > >
> > > Thank you for your thoughtful feedback and for recognizing our contributions. We greatly appreciate your time and effort in reviewing our work and are glad that the additional experiments addressed your concerns.

---

### Author Response · Authors · 2024-11-23

We sincerely thank all the reviewers for their time, effort, and valuable feedback on our paper. We appreciate the constructive comments and insightful questions, which have helped us further clarify and strengthen our work.

We are particularly encouraged by the positive feedback from Reviewers MUMM and h4fS, who both acknowledged the novelty and strengths of our contributions. Notably, Reviewer MUMM updated their score to 8 (accept, good paper), highlighting their appreciation for the refined partitioning approach and the introduction of Refined Pareto Stationarity (RPS) as a sharper solution concept.

In our individual responses, we have addressed questions and concerns raised by the reviewers. We hope these clarifications resolve any ambiguities and further demonstrate the value of our contributions.

We summarize the changes made in the latest revision (highlighted in purple) as follows:
1. Added experiments comparing RP-MGDA and MGDA with *stochastic* gradients to address questions from Reviewers MUMM and Vj3K (Appendix B.3.4).
2. Added experiments comparing PCGrad with and without refined partitioning, further supporting the RP approach and addressing Reviewer EgBd's question (Appendix B.3.5).
3. Included experiments using ELU as the activation function (ensuring differentiability everywhere) to address Reviewer MUMM's question (Appendix B.3.3).
4. Provided pseudocode for the REFINED_PARTITION() procedure in Appendix A.1, offering detailed reference for the process described in the main paper (Lines 285–286).
5. Corrected some notation issues and added further justification to the proof of Theorem 3 (Page 16), addressing Reviewer MUMM's question.

---

### Meta-Review · Area_Chair_gxvh · 2024-12-14

**Metareview:**

This paper studies gradient-based multi-objective optimization (MOO). In particular, the authors first revealed some limitations of the Pareto Stationarity, and proposed a novel Refined Pareto Stationarity (RPS) and the associated RP-MGDA algorithm. RPS provides a sharper characterization than Pareto Stationarity and RP-MGDA is proved to converge to RPS. Numerical experiments demonstrated the advantages of RP-MGDA over the original MGDA. The authors are advised to incorporate various concerns that the reviewers raised into the final version of the paper.

**Additional Comments On Reviewer Discussion:**

Added numerical experiments and clarified some proof steps.

---

### Decision · Program_Chairs · 2025-01-22

Accept (Poster)